

**Global 5-km resolution estimates of secondary evaporation including irrigation through satellite data assimilation**

Albert I.J.M. van Dijk[1], Jaap Schellekens[2,3], Marta Yebra[1], Hylke E. Beck[4], Luigi J. Renzullo[1], Albrecht Weerts[2,5], Gennadii Donchyts[2]

[1] Fenner School of Environment & Society, Australian National University, Canberra, ACT, Australia

[2] Deltares, Delft, The Netherlands

[3] Vandersat B.V., Haarlem, The Netherlands

[4] Princeton University, Princeton, NJ, USA

[5] Wageningen University & Research, Wageningen, The Netherlands



**Abstract**
A portion of globally generated surface and groundwater resources evaporates from wetlands, water
bodies and irrigated areas. This secondary evaporation of 'blue' water directly affects the remaining
water resources available for ecosystems and human use. At the global scale, a lack of detailed water
balance studies and direct observations limits our understanding of the magnitude and spatial and
temporal distribution of secondary evaporation. Here, we propose a methodology to assimilate
satellite-derived information into the landscape hydrological model W3 at an unprecedented 0.05° or
c. 5 km resolution globally. The assimilated data are all derived from MODIS observations, including
surface water extent, surface albedo, vegetation cover, leaf area index, canopy conductance, and land
surface temperature (LST). The information from these products is imparted on the model in a simple
but efficient manner, through a combination of direct insertion of surface water extent, evaporation
flux adjustment based on LST, and parameter nudging for the other observations. The resulting water
balance estimates were evaluated against river basin discharge records and the water balance of closed
basins and demonstrably improved water balance estimates compared to ignoring secondary
evaporation (e.g., bias improved from +38 mm/d to +2 mm/d). The evaporation estimates derived
from assimilation were combined with global mapping of irrigation crops to derive a minimum
estimate of irrigation water requirements ($I_0$), representative of optimal irrigation efficiency. Our $I_0$
estimates were lower than published country-level estimates of irrigation water use produced by
alternative estimation methods, for reasons that are discussed. We estimate that 16% of globally
generated water resources evaporate before reaching the oceans, enhancing total terrestrial
evaporation by $6.1\cdot10^{12}$ $m^3$ $y^{-1}$ or 8.8%. Of this volume, 5% is evaporated from irrigation areas, 58%
from terrestrial water bodies and 37% from other surfaces. Model-data assimilation at even higher
spatial resolutions can achieve a further reduction in uncertainty but will require more accurate and
detailed mapping of surface water dynamics and areas equipped for irrigation.



## Introduction

The generation of surface and groundwater resources is commonly conceptualised one-dimensionally
as the net difference between precipitation, evaporation (including transpiration) and soil storage
change. However, some part of the generated 'blue' water (Falkenmark and Rockström, 2004)
subsequently inundates floodplains, accumulates in wetlands and freshwater bodies, or is extracted for
irrigation. A fraction of that water will evaporate in this second instance. This 'secondary
evaporation' directly reduces the remaining blue water resources available for ecosystems and
economic uses downstream but also increases the use of water by terrestrial ecosystems before
discharging into the oceans. At the global scale, our understanding of the magnitude and
spatiotemporal distribution of secondary evaporation is limited by a lack of detailed water balance
studies and direct observations. Until recently, land surface models ignore lateral water transport and
secondary evaporation altogether or provide a rudimentary description. This is understandable, given
the complexity and computational challenge in simulating the lateral redistribution and secondary
evaporation of water at the global scale. However, it is increasingly clear that the lateral redistribution
of water cannot be ignored in global water resources analyses (Oki and Kanae, 2006; Alcamo et al.,
2003), carbon cycle analysis (Melton et al., 2013) and regional and global climate studies (e.g., Thiery
et al., 2017).
Even approximate numbers on the importance of secondary evaporation in the global water cycle are
not available. Oki and Kanae (2006) derived global bulk estimates of gross evaporation from lakes,
wetlands and irrigation (combined $10.1 \cdot 10^{12}$ m$^3$ y$^{-1}$) but their estimate was based on modelling only
and included both primary and secondary evaporation. There have been some studies estimating
irrigation water requirements at the global scale (Döll and Siebert, 2002; Wada et al., 2014; Siebert
and Döll, 2010) but these studies were based on idealised modelling, did not attempt to separate
between primary and secondary evaporation, and did not consider other sources of secondary
evaporation.
There have been attempts to use satellite observations to estimate the importance of secondary
evaporation at a regional scale. For example, Doody et al. (2017) used MODIS-based evaporation
estimates (Guerschman et al., 2009) over Australia to delineate areas receiving lateral inflows. They
used ancillary data to attribute these to surface water inundation, irrigation, and groundwater-
dependent ecosystems, respectively. At the global scale, Wang-Erlandsson et al. (2016) used satellite-
based ET estimates from several sources to infer rooting depth, which provided some insights into the
spatial distribution of surface- and groundwater dependent ecosystems.
Historically, three contrasting approaches have been followed to estimate evaporation: water balance
modelling; inference from land surface temperature (LST) remote sensing; and estimation based on
vegetation remote sensing. All three approaches rely on meteorological data and effectively involve a
land surface model of some description, albeit of variable complexity. Hybrids between the three
approaches have also been developed over time to mitigate respective weaknesses (Glenn et al.,



2011). For example, dynamic simulation of the soil water balance can provide a valuable constraint on
satellite-based evaporation estimates in water-limited environments; provided precipitation is the only
source of water for evaporation, and accurate precipitation estimates are available (Glenn et al., 2011;
Miralles et al., 2016). However, where there are additional sources of water or unexpected soil
moisture dynamics, applying this constraint can degrade evaporation estimates.
Beyond dynamic hydrological models, evaporation products based more closely on vegetation remote
sensing implicitly account for the effect of lateral water redistribution on transpiration, but often do
not account for open water evaporation (Yebra et al., 2013; Zhang et al., 2016), with exceptions
(Guerschman et al., 2009; Miralles et al., 2016). Satellite-observed LST has a direct, physical
connection to the surface heat balance, and through the overall surface water and energy balance can
provide a constraint on evaporation estimates. Several techniques have been developed to infer
evaporation from LST, and many successful applications at local scale have been documented (Kalma
et al., 2008). Over larger areas, the application of LST-based methods is complicated by the need for
time-of-overpass estimates of radiation components, air temperature, and aerodynamic conductance
(Kalma et al., 2008; Van Niel et al., 2011). There are promising developments that can overcome
some of these challenges (Anderson et al., 2016), although they are yet to be fully evaluated.
Arguably, the most promising approach to evaporation estimation is to combine water balance
modelling, LST remote sensing, and vegetation remote sensing within a model-data fusion
framework. This prospect motivated the present study.
*Aim*
Our objective was to develop a methodology to assimilate optical and thermal observations by the
MODIS satellite instruments into a 0.05° resolution global hydrological model to estimate
evaporation and to evaluate the quality and quantitative accuracy of the resulting estimates as much as
possible. Based on the resulting estimates, we wished to answer the following questions:
• What is the magnitude of secondary evaporation of surface and groundwater resources in the
global and regional water cycle?
• What is the magnitude of irrigation evaporation and how does it relate to total agricultural water
withdrawals?
• What are the contributions of secondary evaporation from irrigation, permanent water bodies,
ephemeral water bodies, and other surfaces?
• Is secondary evaporation likely to have a noticeable impact on the global carbon cycle and
climate system?

**Materials and Methods**
*Global water balance model description*




The World-Wide Water model (W3) version 2 is an evolution of the AWRA-L and W3RA group of
models. The AWRA-L model is used operationally for water balance estimation across Australia at
0.05° resolution by the Bureau of Meteorology. An overview of the operational AWRA-L model
(version 5) can be found in Frost et al. (2016b), with details on the scientific basis in Van Dijk (2010).
Very briefly, the model operates at daily time step and is grid-based. Each cell is conceptualised to
represent several parallel small, identical catchments. The soil column is conceptualised as a three-
layer unsaturated zone overlaying an unconfined groundwater store. The unsaturated soil water
balance and corresponding water and energy fluxes can be simulated separately for hydrological
response units (HRUs) that each occupy a fraction of the grid cell. Sub-grid parameterisations are
applied to simulate the area fractions with surface water, groundwater saturation and root water access
to groundwater dynamically, based on the hypsometric curves (i.e., the cumulative distribution
function of elevation) for each grid cell (Peeters et al., 2013).
The W3 (version 2) model is a global implementation of AWRA-L (version 5) at the same 0.05°
resolution. Important differences are as follows (details in Appendix A). Separate HRUs were not
considered, however, the water balance of permanent water bodies is calculated separately. Global
gridded climate time series and surface, vegetation and soil parameterisation data were used. We used
the cumulative distribution function of Height Above Nearest Drainage (HAND; Nobre et al., 2015)
for each grid cell instead of hypsometric curves, which we derived from high-resolution global digital
elevation models. Five model parameters that were both relatively uncertain and influential were
calibrated and regionalised using large global data sets of site measurements evaporation and near-
surface soil moisture, and a global dataset of catchment streamflow records (the parameters represent
proportional adjustments to initial estimates of, respectively, maximum canopy conductance, relative
canopy rainfall evaporation rate, soil evaporation, saturated soil conductivity, and soil conductivity
decay with depth). Differences less relevant here include the addition of a snow water balance model
and grid-based river routing. A range of W3-simulated water and energy balance terms has been made
publicly available as part of 'Tier-2' of the eartH2Observe project (Schellekens et al., 2017). The
AWRA-L and W3 models have received extensive evaluation, demonstrating realistic estimates of
evaporation, soil moisture, deep drainage, streamflow and total water storage (e.g., for more recent
implementations, Tian et al., 2017; Frost et al., 2016a; Beck et al., 2016; Holgate et al., 2016).
*Data assimilation*
All data assimilated here were derived from NASA's Moderate Resolution Imaging
Spectroradiometer (MODIS) instruments. The data included albedo, reflectance, leaf area index (LAI)
and LST (details in Appendix A). We followed the following steps, except for LST. First, the MODIS
band reflectances were used to estimate vegetation cover fraction and canopy conductance following
Yebra et al. (2015; 2013); surface water extent was estimated following Van Dijk et al. (2016); and
MODIS albedo, snow cover fraction and LAI products were used in their original form. Next, seven
model states were updated using a simple nudging scheme. For each state, the observation and model





error estimates were based on an assessment of the noise in the observational data, the expected
dynamic rate of change, and the expected skill of the model. The resulting 'gain' factors (i.e. the
relative weight of observations) varied from 0.5 for LAI and snow fraction to 0.99 for surface water
fraction. The updated states were also used dynamically to update six related parameters of diagnostic
model equations, including a parameter relating vegetation cover fraction to canopy conductance,
another relating vegetation cover to LAI, and four parameters relating surface state to albedo.
The approach to assimilate LST observations was different. In this case, the dynamic model was run
one timestep forward to produce a background estimate of the surface energy balance and evaporation
flux. The corresponding average daytime LST ($T_s$, K) was estimated from the average daytime
sensible heat flux ($H$, W m$^{-2}$) as
$$T_s = T_a + \frac{H}{\rho_a c_p g_a} \tag{1}$$
where $T_a$ is air temperature (K), $\rho_a$ air density (kg m$^{-3}$), $c_p$ specific heat capacity (J kg$^{-1}$ K$^{-1}$), and $g_a(u)$
aerodynamic conductance (mm s$^{-1}$). The latter is a function of wind speed scaled by the wind speed
measurement and vegetation heights, respectively, following Thom (1975).
Poor characterisation of spatial gradients in radiative exposure, air temperature, and wind speed in
areas with relief can cause a poor relationship between observed and modelled LST (Kalma et al.,
2008). Fortunately, secondary evaporation primarily occurs in regions with low relief. Therefore, data
assimilation was only attempted for areas with an average slope less than 3% (as calculated from the
higher resolution DEM; Appendix A). This threshold was empirically found to include a large
majority of observed surface water inundation and mapped irrigation areas.
A second challenge relates to the inconsistency between the observation time-of-overpass LST and
model-predicted mean daytime LST. We assumed that time-of-overpass and mean daytime LST will
have different spatial averages, but share a near-identical spatial pattern of deviations from the spatial
averages. This assumption also helps to remove systematic bias, which is the largest source of error in
MODIS LST estimates. Previous assessments report errors in MODIS that are within 0.7 K under
conducive atmospheric conditions but can increase to 3 or 4 K due to errors in atmospheric correction
that tend to cause similar level of bias over a larger area (Wan et al., 2004; Wan, 2008; Wan and Li,
2008; Hulley et al., 2012).
In the assimilation step, the median observed and modelled LST were calculated for all low-relief grid
cells within a spatial window of 15° latitude and longitude and subtracted from the respective gridded
LST values. Subsequently, we calculated the difference between resulting observed and modelled
LST values. The calculated difference was reduced by up to 1 K to conservatively allow for
uncertainty in the assumptions and errors in the observations. Next, the model LST was updated with
the remaining difference towards the MODIS-observed LST. An updated latent heat flux ($\lambda E'$ in W m$^-$
$^2$; the prime indicating the updated variable) can be calculated from the energy balance as





$$\lambda E' = A - H' = A - \rho_a c_p g_a (T_s' - T_a) \qquad\qquad (2)$$
where $A$ is available energy (W m$^{-2}$). To ensure physical consistency within the model context, $\lambda E'$
was constrained to positive values below or equal to potential evaporation $E_0$, calculated following
Penman-Monteith theory (details in Van Dijk, 2010). Temporal consistency was ensured by recording
the ratio $\lambda E'/\lambda E$ and using it to adjust simulated $\lambda E$ for subsequent days until a new LST observation
was available. Finally, $E$ was calculated through division by the latent heat of vaporisation $\lambda$.
To illustrate the data assimilation, time series of observations and model results for one 0.05° grid cell
in the Nile delta in Egypt are shown in Figure 1. This grid cell was chosen because it represents one of
comparatively few grid cells worldwide deemed to be 100% equipped for irrigation in global mapping
(although annual maximum NDVI derived from Landsat suggests that only 80–81% of the area is in
fact irrigated; Figure 1a). The processing steps are illustrated by a comparison of observed,
background and analysis LST estimates for the year 2002 (Figure 1b), and the resulting sensible heat
flux (Figure 1c) and daily evaporation (Figure 1d). Corresponding temporal patterns in the
evaporative fraction ($E/E_0$) show that data assimilation brings the temporal pattern of evaporative
fraction in close agreement with satellite-observed vegetation cover fraction (Figure 1e), which
provides as a largely independent consistency test.



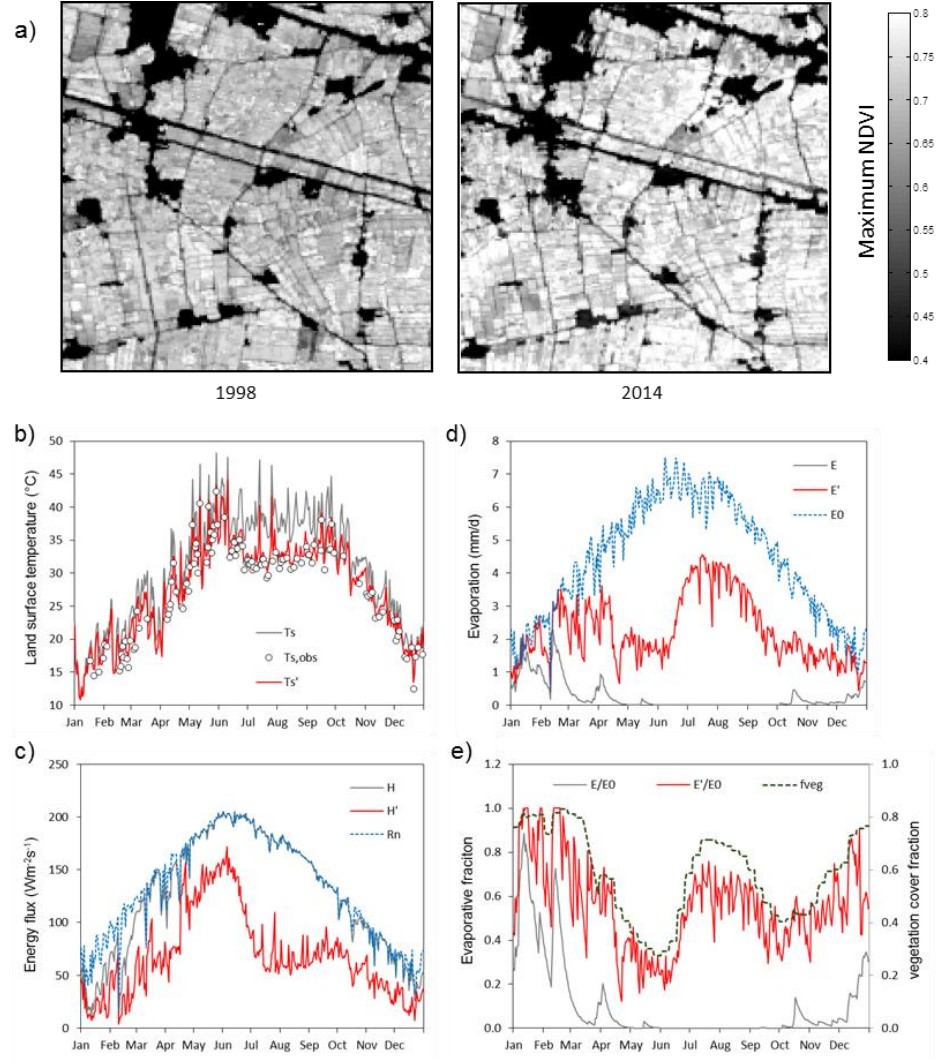

Figure 1. Illustration of method to assimilation MODIS land surface temperature observations. Data shown are for 2002, for 0.05° grid cell in the Nile River delta, Egypt (centred 31.075°N, 30.325°E). (a) *Maximum normalised difference vegetation index* (NDVI) derived from Landsat imagery provided by Google Earth Engine, suggesting that effectively 81% and 80% of the grid cell was cropped in 1998 and 2014, respectively. (b) *Land surface temperature*: background ($T_s$, grey line), observed ($T_{s,obs}$, circles) and analysis ($T_s'$, red line) estimates for the grid cell with average bias across the 15° window removed. (c) *Sensible heat flux*: background ($H$, grey) and analysis ($H'$, red) estimates along with net radiation ($R_n$, blue). (d) *Evaporation*: background ($E$, grey) and analysis ($E'$, red) estimates along with potential evaporation ($E_0$, blue). (e) *Evaporative fraction*: background ($E/E_0$, grey) and analysis ($E'/E_0$, red) along with vegetation cover fraction derived from MODIS NDVI ($f_{veg}$, green).



*Irrigation water use estimation*

For irrigated areas, the long-term average difference between precipitation and total evaporation derived from data assimilation provides an estimate of the importance of additional water inputs. However, it cannot be interpreted directly as an estimate of irrigation water requirements, much less as an estimate of water withdrawals. This is because precipitation and crop water requirements are both unevenly distributed in time, and there is limited water storage capacity in the crop root zone. Additional water is lost from the root zone through drainage and runoff, which will need to be compensated by additional irrigation inputs. This field-level irrigation inefficiency does not necessarily change the long-term net water balance: provided total precipitation and evaporation do not change, the additional inputs will equal the additional runoff and drainage. However, such inefficiencies do need to be accounted for when estimating the total amount of irrigation water required (Siebert and Döll, 2010).

Estimating total field-level irrigation water requirements is sensitive to assumptions about the capacity for added water to remain stored in the root zone irrigation and about strategies (e.g., pursuing a stable low or high soil moisture or paddy water level, suboptimal or soil moisture deficit irrigation, flood irrigation or partial drip irrigation, and so on). Here, we estimated a minimum field-level irrigation requirement ($I_0$ in mm), which can be taken as a conservatively low estimate of irrigation that represents highly efficient irrigation practices.

We used global mapping by crop type to estimate $I_0$ using a plausible range of published assumptions about water storage capacity. It was assumed that irrigation is just sufficient to replenish lost water without any direct drainage or runoff losses; that is, losses only occur when precipitation exceeds available storage capacity. Following Siebert and Döll (2010), we estimate the available root zone storage ($S_{max}$ in mm) capacity for $i=1..26$ irrigated crop types based on the estimated harvested area ($A_i$ in ha) of each as contained in the MIRCA2000 dataset (Portmann et al., 2010). These numbers are combined with assumed rooting depth ($z_i$) and the allowable fraction of depletion of available soil water $p_i$ (Allen et al., 1998) for each crop type as proposed by Siebert and Döll (2010). The plant available water content ($\theta_a$) was estimated using global soil property data (Shangguan et al., 2014; see Appendix A), calculated as the difference between $\theta$ at field capacity and permanent wilting point, assumed to correspond to water potential values of -3.3 and -150 m, respectively. In formula:

$$S_{max} = \frac{\sum A_i z_i p_i}{\sum A_i} \theta_a f_{irr} \qquad (3)$$

where $f_{irr}$ is the fraction of the grid cell area that is equipped for irrigation (Portmann et al., 2010). This method produced a global average root zone storage of 51 mm per unit of irrigated land, with 90% of values between 10–85 mm, with values depending primarily on the value of $z_i$.

Because we have observation-based estimates of evaporation, we do not simulate the influence of soil water status on evaporation, but instead, propagate a simple water balance model forced with



evaporation estimates. In words, the change in soil moisture storage from one day ($S_t$) to the next
($S_{t+1}$) is the net result of gross rainfall onto the irrigated area ($P_{irr}$), evaporation from the irrigated area
($E_{irr}$), the minimum irrigation water application required ($I_0$) and drainage ($D$), with storage and
cumulative fluxes (all in mm):
$$S_{t+1} = S_t + P_{irr} - E_{irr} + I_0 - D \tag{4a}$$
Partial rainfall ($P_{irr}$) is proportional to the irrigation fraction:
$$P_{irr} = f_{irr}P \tag{4b}$$
It is assumed that any increase in the estimate of evaporation ($E'–E$) from data assimilation is due to
irrigation, where this occurs, and therefore $E_{irr}$ is given by:
$$E_{irr} = f_{irr}E + (E' - E) \tag{4c}$$
Any soil water additions more than maximum storage capacity ($S_{max}$) are assumed to become
drainage, and irrigation is assumed to be just enough to prevent $S<0$:
$$I_0 = max(E_t - P_g - S_t, 0) \tag{4d}$$
$$D = max(S_t + P_g - E_t - S_{max}, 0) \tag{4e}$$
Rainfall interception losses are included in $E$. Surface runoff and residual drainage are assumed
negligible when $S<S_{max}$. This is an important simplification, but consistent with the definition of a
minimum irrigation requirement estimate that reflects optimal efficiency. The daily water balance
model was evaluated with an initial state of $S=S_{max}$ and propagated from 2000−2014. The first year
was not used in subsequent calculations to allow for artefacts from the initial state chosen.
*Evaluation of basin water balance*
One test of the accuracy of secondary evaporation estimates is to evaluate whether their inclusion in
the basin water balance improves agreement with observations. The difference between $E'$ derived
from data assimilation and the background estimate $E$ is interpreted to be derived from lateral inflows:
$$E_{lat} = E' - E \tag{5a}$$
For any basin, the total net amount of discharge from the basin ($Q_n$) is the result of the gross amount
of streamflow generated in all tributaries ($Q_g$) minus secondary evaporation of flows downstream
($E_{lat}$) and the change in storage derived from those flows ($\Delta S_{lat}$):
$$Q_n = Q_g - E_{lat} - \Delta S_{lat} \tag{5b}$$
Natural storage variations in soil and groundwater and river channel storage are explicitly simulated
by the model and not included in $\Delta S_{lat}$. Storage changes in other surface water bodies (e.g., lakes and
reservoirs), river-groundwater exchanges, and induced soil or groundwater storage changes directly
related to inundation or irrigation (including pumping) would affect $\Delta S_{lat}$. It is assumed here that the





magnitude of $\Delta S_{lat}$ is negligible compared to the other terms if fluxes are averaged over the period
2001–2014. This needs to be considered when interpreting results for individual basins.
We used discharge data for large basins to evaluate whether our estimates of $E_{lat}$ improved the overall
agreement between modelled and observed $Q_n$. The river discharge data used were drawn from the
global database of end-of-river discharge records compiled by Dai et al. (2009). This includes data for
925 rivers worldwide. Out of these, we considered only basins for which more than five years of data
were available during 1995–2014. This longer period was adopted because few basins had sufficient
measurements after 2000. To avoid errors arising from differences in the delineation of basins, we
rejected basins with a catchment area less than 100,000 km$^2$ and those with a reported drainage area
that was more than 25% different from the DEM-derived basin area at the river mouth. For the
remaining 38 large basins, the temporal and area-average discharge was calculated and compared to
the modelled $Q_n$ and $Q_g$ (all in mm y$^{-1}$).
Closed or endorheic basins represent a special case where $Q_n$=0 and can also be used to construct a
water balance. The 0.05° flow direction grid was used to delineate all internally draining basins
located between 72°N and 60°S (further poleward the DEM is affected by land ice). Adjoining
endorheic basins were merged into contiguous regions to avoid incorrect basin delineation. From the
resulting regions, all those with a surface area greater than 50,000 km$^2$ were extracted, resulting in 13
contiguous regions. For these regions, Eq. (5b) was evaluated and compared to the expected $Q_n$=0.
The LST data assimilation changes evaporation without adjusting other water balance terms and
hence does not conserve mass balance. In both open and closed basins, this can produce a positive or
negative $Q_n$ from Eq. (5b). A difference between estimated and observed $Q_n$ can occur for any of four
reasons: $Q_g$ is underestimated, $E_{lat}$ overestimated, $\Delta S_{lat}$ is non-negligible, or (for discharging basins
only) recorded $Q_n$ is in error.
*Evaluation of apparent irrigation water use*
Evaluating estimates of secondary evaporation due to irrigation is challenging. Direct observations of
evaporation from irrigated land are not widely available, represent point observations, and include
primary evaporation. At basin or country level, estimates of irrigation water use can be categorised as
'bottom-up' or 'top-down' estimates. Bottom-up estimates require scaling of estimated crop water use
to field-level irrigation requirements. Top-down estimates involve estimating large-scale withdrawals
(e.g., by differencing of discharge measurements along a river reach or measured bulk diversions) and
accounting for "project" or scheme losses along the distribution network (Bos and Nugteren, 1990).
Both approaches have large uncertainties but provide estimates of the order of magnitude of irrigation
water use.
Bottom-up estimates of irrigation water use at the global scale and for individual countries are
available from previous studies (Siebert et al., 2010; Wada et al., 2014; Siebert and Döll, 2010). They
involve soil-vegetation water balance modelling. Similar to the approach used here, these methods





require assumptions about root zone storage capacity, the rate of drainage of water from the root zone,
the permissible range of root zone soil moisture, and the efficiency of irrigation. Unlike the approach
used here, they furthermore require assumptions about evaporation, usually following FAO's crop
factor approach (Allen et al., 1998) to model crop water use. The resulting one-dimensional irrigation
water requirement estimates are subsequently extrapolated spatially using mapping of areas equipped
for irrigation (e.g., Portmann et al., 2010), using assumptions about the number of crop rotations and
the area factually irrigated. Each of these assumptions introduces errors and uncertainties.
Nonetheless, a comparison with these studies should provide insight into the method developed here.
An important source of uncertainty in our estimation of large-scale $I_0$ is due to the diffuse spatial
distribution of irrigated areas, which is further amplified in current mapping products. The mapping of
areas equipped for irrigation contained in the MIRCA2000 dataset (Portmann et al., 2010) was done at
0.08° grid resolution and linearly interpolated to 0.05° resolution in this study. Even at this high
resolution, a large proportion of total irrigable land occupies only a small fraction of a grid cell
(Figure 2).

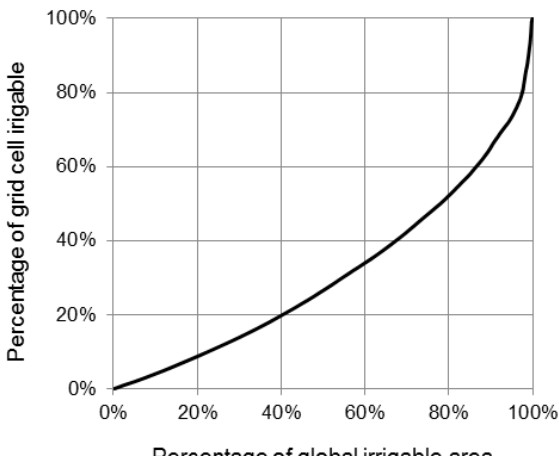


Figure 2. Cumulative distribution curve or quantile plot describing the degree to which the global irrigable area
is concentrated. It shows that, at 0.05° grid resolution, almost half of the total global irrigable area occupies less
than 25% of a grid cell.

The degree of concentration differs between countries for two reasons. Firstly, the true distribution of
irrigation land varies; for example, irrigation tends to be highly concentrated in large surface water
irrigation schemes (e.g., the Nile delta and Indus floodplains) but can be highly distributed where
supplementary irrigation water is drawn from unregulated streams or groundwater. Secondly, the


quality, resolution and predictive value of information related to irrigation area varies widely, which
affects the accuracy of mapping (Portmann et al., 2010). The distribution of irrigation land introduces
uncertainty in the attribution of $E'$ in grid cells with small fractions of irrigated land. We expect that
the fraction of a grid cell that needs to be irrigated to create a measurable LST signal may be around
10% but will vary spatially depending on the LST contrast between irrigated and non-irrigated land.
To account for this uncertainty, we calculated the mean $I_0$ (Eq. 4) per unit irrigation area for all grid
cells with more than, respectively, 1, 2, 5, 10 and 25% of the area equipped for irrigation. These
estimates were subsequently multiplied with the total area equipped for irrigation in each country. The
coefficient of variation among the five estimates was calculated as a measure of estimation
uncertainty.
The AQUASTAT database (FAO, 2017) provides country-level estimates of agricultural water
withdrawal ($W$ in km$^3$ y$^{-1}$) from surface and groundwater. The estimates are derived by different
methods for different countries, and likely include both bottom-up and top-down techniques.
Estimates also relate to different periods or years. Despite these uncertainties, they currently represent
official international statistics for each country. Any comparison of field-level irrigation water
application ($I_0$) and large-scale water withdrawal ($W$) needs to account for inefficiencies in the entire
water distribution network. These include evaporation, leakage and return flow on- and off-farm.
'Project efficiencies' that express the ratio of $I_0$ over $W$ can be estimated in principle, but this requires
detailed ancillary data (Bos and Nugteren, 1990). In their global modelling study, Siebert and Döll
(2010) proposed ratios range from 0.25 for irrigation dominated by paddy rice to 0.70 for efficient
crop irrigation methods in Canada, Northern Africa and Oceania. We did not assume values but
instead calculated an 'apparent' bulk project efficiency for each country, by dividing the ratio of
modelled $I_0$ over $W$ reported in AQUASTAT. The credibility of the resulting values was subsequently
interpreted within the framework developed by Bos and Nugteren (1990).
*Secondary evaporation and the global water cycle*
Total secondary evaporation was estimated as the sum of open water evaporation plus the difference
$E'-E$, representing the difference between modelled primary evaporation $E$ for a situation where
precipitation is the only source of water (the background estimate) and total evaporation $E'$ resulting
from LST assimilation (the analysis estimate). The resulting estimate of total secondary evaporation is
a hypothetical and model-based quantity. Evaporation in the absence of lateral flows is counterfactual
and not necessarily accurately estimated by the model, particularly in humid environments.
Furthermore, all open water evaporation was included in secondary evaporation; we did not attempt to
estimate the evaporation that might have occurred from the surface had it not been covered by water.
The difference $E'-E$ was distributed dynamically in proportion to the magnitude of each of three
evaporation terms (i.e., transpiration, soil evaporation, and open water evaporation; wet canopy
evaporation was left unchanged). A component of secondary evaporation was attributed to irrigation
following the method described earlier. The remainder could be attributed to permanent water bodies,



ephemeral water bodies, and a residual component that includes any evaporation from replenished
wetlands and floodplains, as well as any use of groundwater sources beyond that simulated by the
model to occur from shallow groundwater (Peeters et al., 2013).

**Results**
*Basin water balance*
The combined surface area of the 51 basins used in evaluation (38 ocean-draining and 13 closed
basins) was 63 million km$^2$ or 47% of the ice-free land surface area (Figure 3). For each region, the
period-average measured discharge (zero in the case of closed basins) was compared with modelled
$Q_g$ and $Q_n$ (Figure 4, Table 1). Overall, accounting for secondary evaporation produced a very small
improvement in the correlation between observed and estimated discharge (Figure 4ab). However, the
largest error contribution was from basins with high discharge rates, where secondary evaporation
represents a small fraction of $Q_g$. A clearer improvement in the agreement was found for basins with
less than 300 mm y$^{-1}$ net discharge (Figure 4cd). The explained variance ($R^2$) increased from 0.67 to
0.71, and there was a reduction of the bias from +38 to +2 mm y$^{-1}$. Water balance estimates were
improved considerably for several basins, including the Indus River ('I' in Figure 4cd), Nile River,
the Great Basin in the USA, and the African Rift Valley (Table 1). The agreement could not improve
where $Q_g$ estimates were already lower than observed, such as the Paraná and Fitzroy Rivers ('P' and
'F' in Figure 4cd). Water balance estimates for some closed basins were also degraded, evident from
negative $Q_n$ values (e.g., the South Interior and Rukwa basins in Southern Africa), implying that $Q_g$
was underestimated, secondary evaporation overestimated, or both (Table 1).

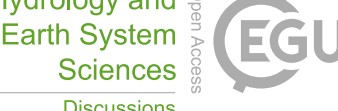

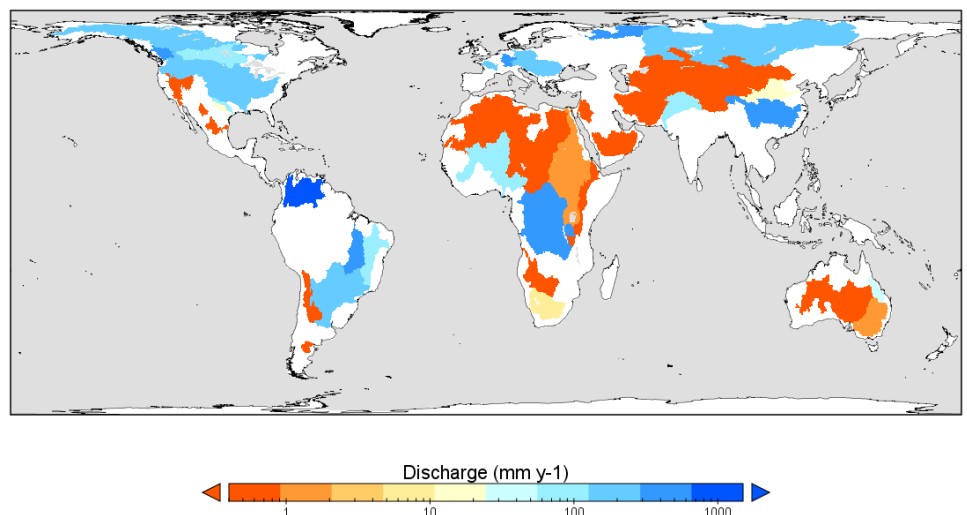


Figure 3. Extent and area-average annual discharge for the 38 ocean-draining (orange to blue) and 13 closed
basins (dark orange) used in the evaluation. The two darkest blue colours indicate a discharge in excess of 300
mm y$^{-1}$.



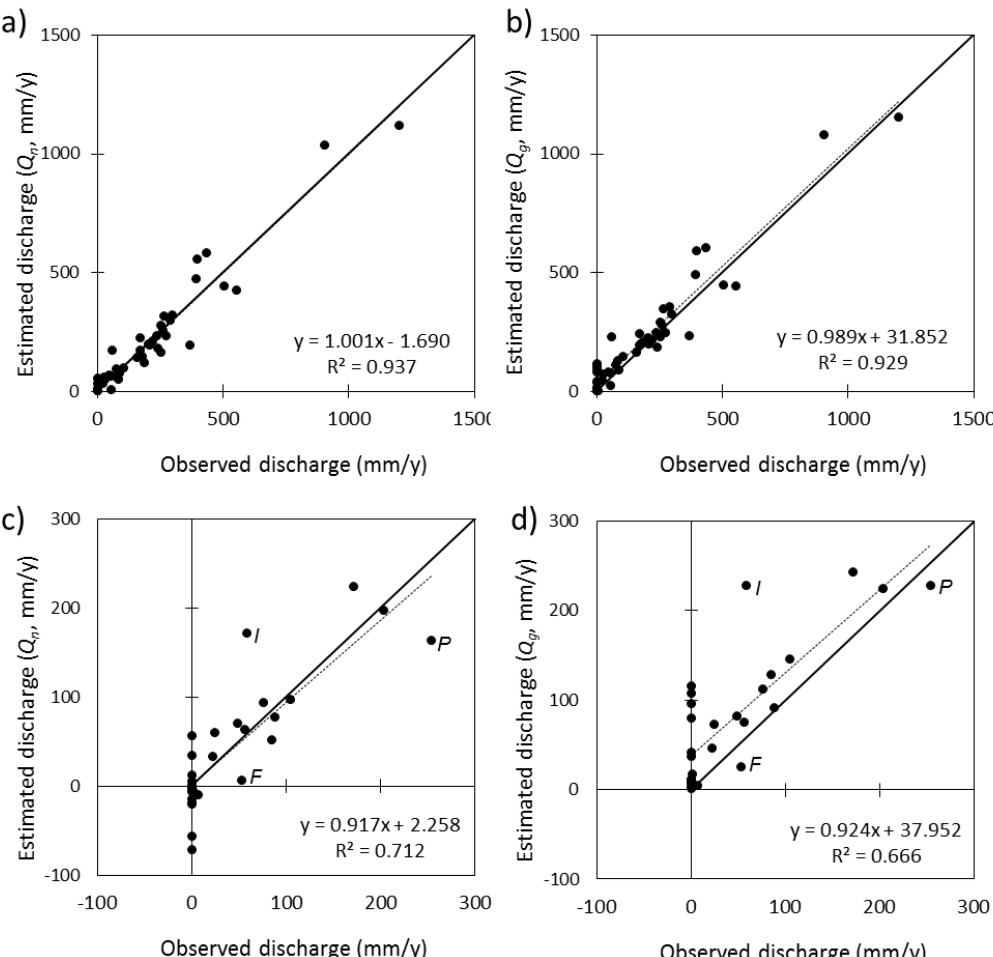


Figure 4. Comparison of observed basin-average discharge (mm y$^{-1}$) for large basins that are internally draining

(i.e., zero discharge) or have adequate station discharge data with model estimates of (a) net discharge ($Q_n$), that

is, gross discharge ($Q_g$) minus secondary evaporation, and (b) $Q_g$ only. (c) and (d) data for discharge below 300

mm y$^{-1}$ only (cf. Table 1). Letters indicate Indus (I), Paraná (P), and Fitzroy (F) River.






Table 1. Area-average discharge (mm y$^{-1}$) for selected basins as observed and estimated by the model in the
presence ($Q_n$) and absence ($Q_g$) of secondary evaporation, respectively. Listed data for basins with discharge
less than 300 mm y$^{-1}$ only (cf. Figure 4cd).

| Area-average basin discharge (mm y$^{-1}$) | | estimated | |
|---|---|---|---|
| | Observed | $Q_n$ | $Q_g$ |
| *Closed river basins* | | | |
| Great Basin, US | - | 1 | 42 |
| Guzman, North America | - | -6 | 3 |
| Mairan-Viesca, Mexico | - | -15 | 7 |
| Patagonia, South America | - | 5 | 10 |
| Titicaca-Chiquita, South America | - | -19 | 38 |
| North Interior, Africa | - | -4 | 4 |
| South Interior, Africa | - | -71 | 12 |
| Rukwa, Africa | - | -56 | 115 |
| Rift Valley, Africa | - | 35 | 107 |
| Jordan | - | -1 | 8 |
| Arabian peninsula | - | 0 | 1 |
| Central Asia | - | 57 | 80 |
| Central Australia | - | -20 | 8 |
| *Ocean-reaching rivers* | | | |
| Nile, Africa | 0 | 13 | 96 |
| Murray, Australia | 1 | -5 | 17 |
| Orange/Senqu, Africa | 7 | -9 | 4 |
| Colorado, US | 23 | 33 | 46 |
| Huanghe, China | 24 | 61 | 73 |
| Burdekin, Australia | 48 | 70 | 82 |
| Parnaiba, Brazil | 76 | 94 | 113 |
| Brazos, US | 57 | 64 | 76 |
| Fitzroy, Australia | 54 | 6 | 26 |
| Indus, Asia | 58 | 172 | 228 |
| Sao Francisco, Brazil | 105 | 97 | 146 |
| Niger/Issa Ber, Africa | 88 | 78 | 92 |
| Nelson, Canada | 85 | 52 | 129 |
| Paraná, South America | 255 | 163 | 228 |
| Elbe/Labe, Europe | 172 | 224 | 243 |
| Mississippi, US | 204 | 198 | 225 |






*Irrigation water requirements*
Spatiotemporal estimates of $I_0$ at 0.05° and daily time step were aggregated to country-level estimates
in km$^3$ y$^{-1}$ (Table 2). Also calculated were the coefficient of variation in $I_0$ estimates ($CV_{I0}$) caused by
the treatment of 'mixed pixels' in irrigation mapping, FAO-reported annual $W$, and the apparent
project irrigation efficiency. Global $I_0$ for 2001–2014 was 680 km$^3$ y$^{-1}$ (standard deviation 110 km$^3$ y$^-$
$^1$). This value is lower than estimates of contemporary irrigation water use reported in the literature of
1092 km$^3$ y$^{-1}$ (Döll and Siebert, 2002), 1180 km$^3$ y$^{-1}$ (Siebert and Döll, 2010) and 994–1179 km$^3$ y$^{-1}$
(Wada et al., 2014). Estimates of $I_0$ listed for seven countries by Döll and Siebert (2002) were all
higher than those found here (Table 2), and even more than double for the USA (112 vs. 48 km$^3$ y$^{-1}$)
and Spain (21 vs 5.1 km$^3$ y$^{-1}$). Quoted independent estimates were 113 km$^3$ y$^{-1}$ for the USA (Solley et
al., 1998) and 15 km$^3$ y$^{-1}$ for Spain (J.A. Ortiz cited in Döll and Siebert, 2002).

Table 2. Irrigation water withdrawal ($W$) as reported to FAO for the 20 countries with largest agricultural
withdrawals, along with the estimated minimum field-level irrigation requirement ($I_0$), the coefficient of
variation in $I_0$ estimates ($CV_{I0}$) and the apparent project efficiency ($I_0 / W$).

| Country | $W$ | $I_0$ | $CV_{I0}$ | $I_0 / W$ |
|---|---|---|---|---|
| | km$^3$ y$^{-1}$ | km$^3$ y$^{-1}$ | - | - |
| India | 688 | 152 | 0.07 | 0.22 |
| China | 392 | 105 | 0.13 | 0.27 |
| United States of America | 175 | 48 | 0.20 | 0.27 |
| Pakistan | 172 | 49 | 0.01 | 0.28 |
| Indonesia | 93 | 14 | 0.10 | 0.15 |
| Iran | 86 | 5 | 0.22 | 0.06 |
| Viet Nam | 78 | 15 | 0.05 | 0.19 |
| Philippines | 67 | 5 | 0.16 | 0.07 |
| Egypt | 67 | 30 | 0.02 | 0.44 |
| Mexico | 62 | 19 | 0.22 | 0.31 |
| Japan | 54 | 4 | 0.23 | 0.07 |
| Iraq | 52 | 5 | 0.19 | 0.10 |
| Thailand | 52 | 16 | 0.09 | 0.32 |
| Uzbekistan | 50 | 11 | 0.02 | 0.21 |
| Brazil | 45 | 16 | 0.39 | 0.36 |
| Turkey | 34 | 6 | 0.36 | 0.16 |
| Bangladesh | 32 | 20 | 0.08 | 0.63 |
| Burma | 30 | 13 | 0.21 | 0.43 |
| Chile | 29 | 2 | 0.22 | 0.07 |
| Argentina | 28 | 5 | 0.47 | 0.17 |
| **Global** | **2,767** | **680** | **0.16** | **0.25** |



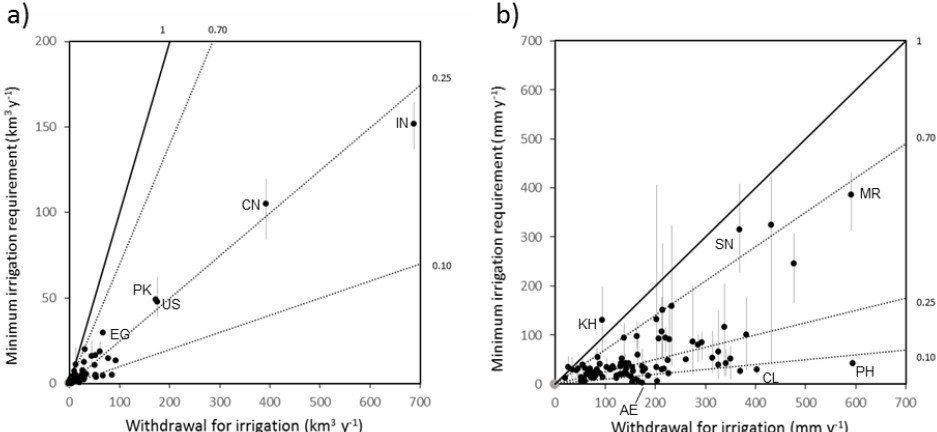


Figure 5. Comparison of country-level agricultural water withdrawal ($W$) (FAO, 2017) and estimated minimum

irrigation requirement ($I_0$) expressed as (a) total volume, and (b) depth per unit area of area equipped for

irrigation for countries with >1 km$^3$ y$^{-1}$ withdrawals ($N$=91). Dotted lines show apparent project efficiencies

between the two quantities. Countries indicated are (in a) Egypt (EG), Pakistan (PK), United States (US), China

(CN) and India (IN), and (in b) Cambodia (KH), Senegal (SN), Mauritania (MR), United Arab Emirates (AE),

Chile (CL), and the Philippines (PH).

434

The $I_0$ explains 96% in the variance in $W$ by country (Figure 5a), but total variance is dominated by

only four countries, and the area equipped for irrigation explains already explains 86% of the

variance. Volumes were divided by the total area equipped for irrigation to normalise for these effects.

Normalised $I_0$ explained 38% of the variance in normalised $W$ (Figure 5b). A high correlation between

the two is not necessarily to be expected, as country-average project efficiencies will vary

(represented by the lines in Figure 5b). For example, a low efficiency is inferred and would be

expected in the Philippines, where irrigation is dominated by paddy rice agriculture, whereas higher

efficiencies would be expected in large schemes in arid countries such as Egypt and Mauritania.

Nonetheless, apparent efficiencies are generally lower than would be expected based on benchmark

estimates provided by Bos and Nugteren (1990). For example, using global volumes of $I_0$ and $W$, a

project efficiency of 0.25 is calculated. This is lower than estimates of 0.36–0.43 assumed in previous

studies (Döll and Siebert, 2002; Wada et al., 2014; Siebert and Döll, 2010). Physically impossible or

implausible project efficiencies were also calculated for some countries, including Cambodia ($I_0/W$

>1), and the United Arab Emirates and Chile ($I_0/W$<0.1) (Figure 5b). Possible explanations for this

will be discussed.






*Secondary evaporation and the global water cycle*
We estimate that secondary evaporation contributed 41.2 mm $y^{-1}$ or 8.1% to total evaporation from the
global land area during 2001−2014 (Table 3), equivalent to 5.4% of terrestrial precipitation (759 mm
$y^{-1}$) and 16% of generated streamflow (258 mm $y^{-1}$). Globally, only a very small percentage of all
secondary evaporation (5%) was due to irrigation. Overall more important pathways for secondary
evaporation were evaporation from permanent water bodies (48%), enhanced transpiration associated
with wetland vegetation or greater-than-predicted groundwater uptake (27%), enhanced soil
evaporation (11%), and evaporation from ephemeral water bodies (10%). Surface and groundwater
inputs enhance global plant transpiration by an estimated 12.1 mm $y^{-1}$, representing a 4.4% increase.
Of this increase, 10% can be attributed to irrigation.

Table 3. Estimates of annual primary and secondary evaporation ($E$ in mm $y^{-1}$) components for 2001-
−2014 expressed as water depths across the global terrestrial area (149·$10^6$ km$^2$).

|  | Primary E | Secondary E | *Total* | Irrigation only |
|---|---|---|---|---|
| wet canopy E | 81.3 | – | **81.3** | – |
| transpiration | 278.7 | 12.1 | **290.8** | 1.2 |
| soil E | 107.0 | 4.9 | **111.9** | 0.5 |
| E from ephemeral water | – | 4.6 | ***4.6*** | 0.3 |
| E from permanent water | – | 19.6 | **19.6** | – |
| *Total* | *467.0* | *41.2* | *508.2* | *2.0* |


The spatial distribution of evaporation from irrigation areas (Figure 6a) and permanent water bodies
(Figure 6b) largely reflects the irrigation and water mapping input data, respectively. The spatial
distribution of other sources of secondary evaporation provides some new insights (Figure 6c).
Globally, some areas with the greatest secondary evaporation volumes include receiving floodplains
in tropical monsoonal regions. The main regions in South America include the Gran Chaco and
Pantanal plains and Amazon floodplains (Figure 7). The main regions in Africa the Southern Interior
basin in Botswana and surrounding countries (including the Okavango Delta and other wetlands), and
the floodplains of the White Nile River in South Sudan and the Inner Niger Delta (Figure 8). Other
areas with high secondary evaporation rates include the Yucatan peninsula in Mexico (Figure 7), the
boreal wetlands and ephemeral lakes of Canada and Scandinavia (Figure 7 and Figure 8,
respectively), and the salt lakes and floodplains of inland Australia (Figure 9).






Figure 6. Spatial distribution of estimated secondary evaporation losses derived from (a) irrigation, (b)

permanent water bodies, and (c) other sources, including wetlands and floodplains.





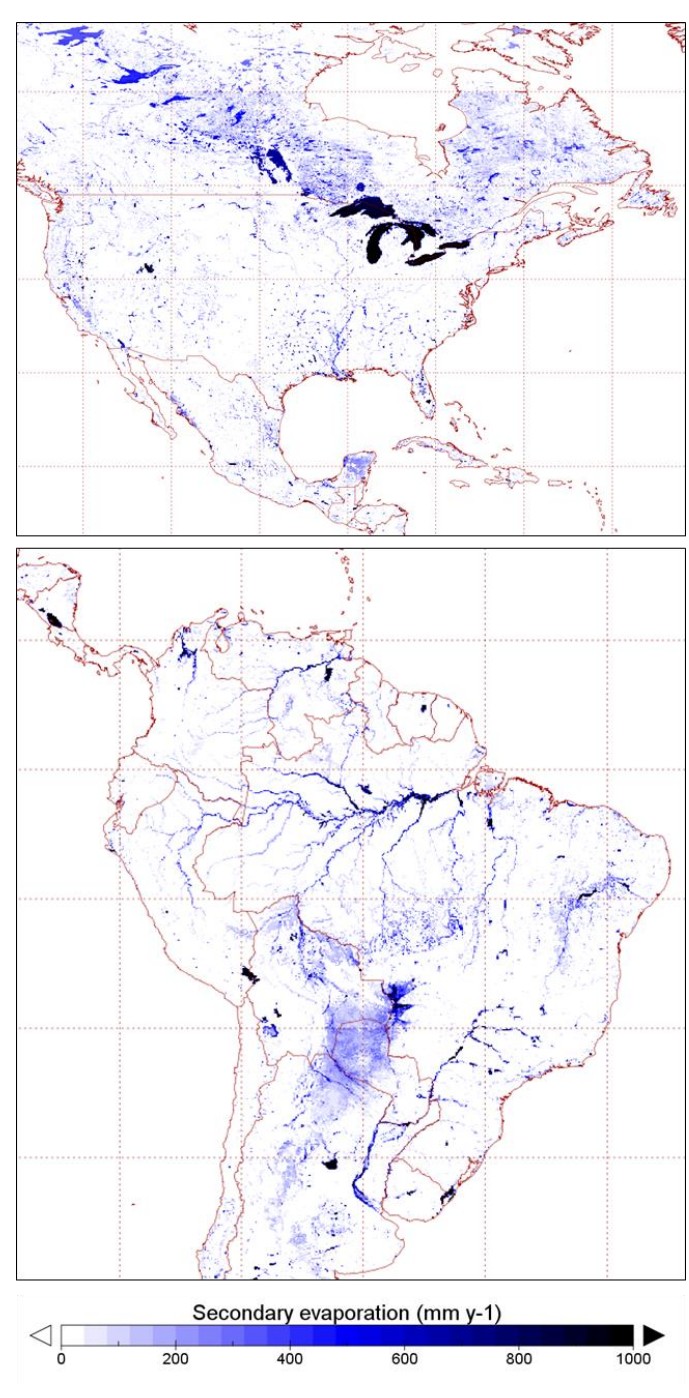


Figure 7. Spatial distribution of secondary evaporation losses in the Americas.





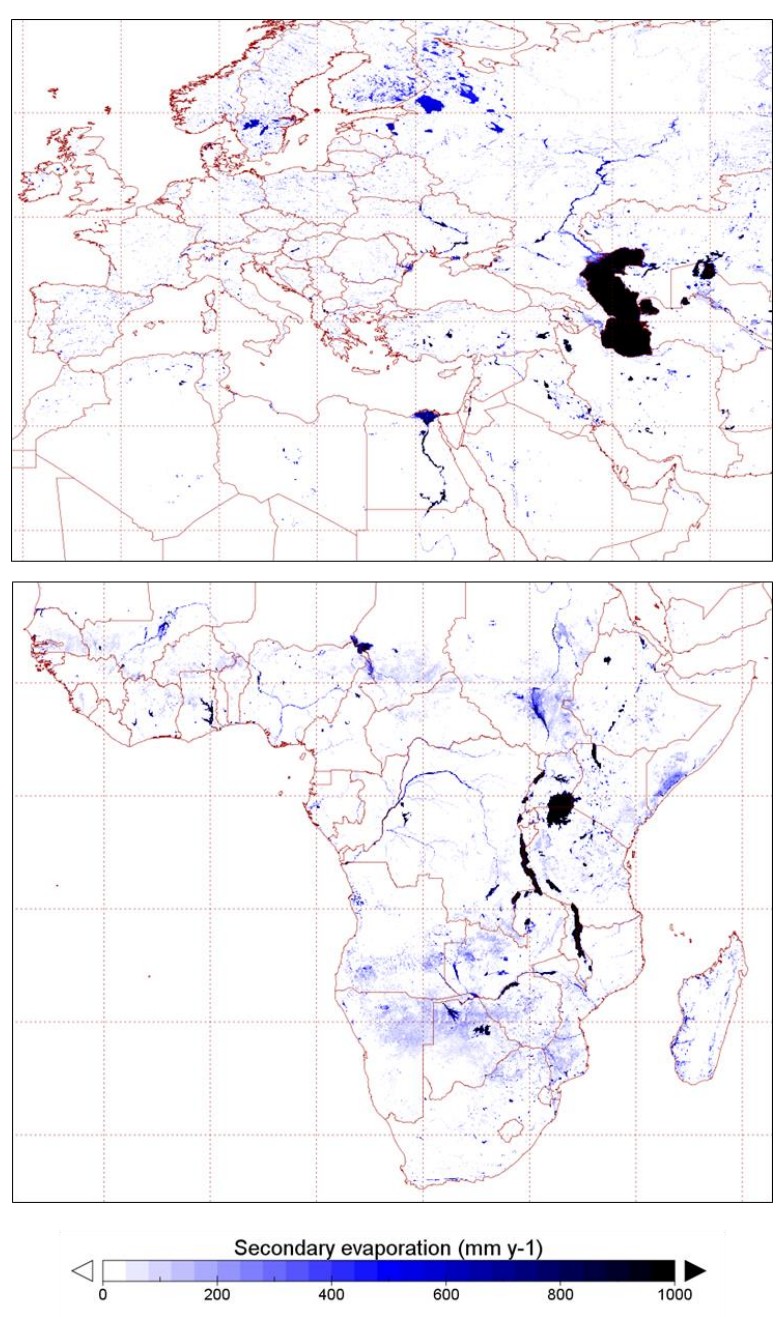


Figure 8. Spatial distribution of secondary evaporation losses in Eurasia and Africa.






Figure 9. Spatial distribution of secondary evaporation losses in Eastern Asia and Oceania.




**Discussion**

*Uncertainties in evaporation estimation*

The uncertainty in estimates of secondary evaporation arises from three main sources: (1) estimation of 'background' evaporation $E$; (2) estimation of surface water evaporation; and (3) estimation of total evaporation $E'$ by LST assimilation. A formal assessment of error in each of these terms is not possible for lack of observations and will vary in space and time. Below we discuss what we expect to be the main sources of uncertainty in each component.

An error in background model $E$ may be compensated by data assimilation, but still leads to an error in the estimated secondary evaporation, calculated as $E'–E$. The main sources of error in $E$ vary as a function of environmental conditions and the quality of the measurement network. In water-limited environments, the most likely sources of error in $E$ are errors in precipitation estimates and the simulation of water availability in the root zone. The quality of precipitation estimates is relatively poor in many of the world's dry regions (Beck et al., 2017). Information on the ability of vegetation to access deeper soil moisture and groundwater is important, particularly in ephemerally wet systems, but is not available at the global scale. In humid environments, the most likely sources of error in $E$ are in the estimation of rainfall interception losses, the net available energy for evaporation, and surface conductance. As part of earlier model development, background $E$ was compared with estimates derived from flux tower observations and compared with alternative ET estimation methods (Yebra et al., 2013; Van Dijk, unpublished). These evaluations showed little if any systematic bias and a standard difference of 135–168 mm y$^{-1}$ across sites ($N$=16–168). This total difference also includes errors in the flux tower-derived estimates and differences arising because the tower footprint is not representative of the grid cell. Therefore the true error in our estimates will be lower.

Observation-based estimates of large-area evaporation from water bodies, wetlands and irrigated areas (i.e. >0.05°) are scarce. Some site measurements of wetland and irrigation evaporation have been published (e.g., Guerschman et al., 2009) but typically reflect an environment with very high spatial variation and therefore often cannot easily be compared to estimates at 0.05°. A coordinated effort that collates observations of secondary evaporation and combines these with historical time series remote sensing imagery (cf. Figure 1a) to generate estimates at a more representative spatial scale would appear necessary and valuable.

Errors in the estimation of surface water evaporation are the combined result of errors in the estimation of open water evaporation rate and the mapping of surface water extent. Open water evaporation rate was estimated using the Priestley and Taylor (1972) approach. An important uncertainty in this approach is that it does not account for strong contrasts in near-surface water temperature. Surface water extent was mapped using 8-day MODIS shortwave infrared (SWIR) reflectance composites (Van Dijk et al., 2016). Systematic overestimation of water extent can occur in low relief regions with very low SWIR reflectance (e.g., lava outflows), whereas underestimation can





occur in regions with a dense elevated canopy that prevents water detection (e.g., floodplain forests or
mature flooded crops).
The LST assimilation mitigates estimation errors in background and open water evaporation but is
also subject to uncertainties of its own. The technique developed here relies on the assumption that
there is a perfect correlation between spatial LST anomalies at the time-of-overpass (around 10 am
local time) and daytime (sunrise-sunset) average values, or at least for the low-relief areas where LST
was assimilated. In reality, there can be spatial differences in the temporal rate of LST change, for
example as a function of spatial differences in heat storage capacity and aerodynamic conductance
(Kalma et al., 2008). Furthermore, we assumed a constant, maximum bias-adjusted error of 1K in the
difference between observed and model background LST. Each of these choices can have affected the
efficacy of the assimilation.
Nonetheless, assessment of temporal patterns in $E'$ (such as in Figure 1e) and the spatial patterns in
secondary evaporation (Figures 6–9) agree with known areas receiving lateral inflows (e.g., wetlands)
or irrigation. Less expected were the widespread high secondary evaporation rates in the northern
Yucatan peninsula in Mexico and the Southern Interior in Southern Africa. The northern Yucatan
peninsula is a low lying region with karst geology and forest are known to access shallow
groundwater (Bauer-Gottwein et al., 2011). The Southern Interior includes several terminal wetlands
(e.g., the Okavango Delta) and has unconsolidated alluvial deposits that contain productive aquifers
(MacDonald et al., 2012) and it is plausible that at least some of the vegetation has access to deeper
soil moisture or groundwater. In both cases, the background evaporation estimate ($E$) is constrained
by precipitation and the corresponding simulated presence of soil- and groundwater within the root
zone ($E$). Any underestimation of $E$ leads to an increased estimate $E'–E$ and therefore an increased
estimate of secondary evaporation, without necessarily implying that all the water involved is derived
from later inflows. An alternative measure of the importance of secondary evaporation is $E'–P$ (Figure
10). These results suggest that period-average $E'$ exceeds $P$ by in the order of 100 to 200 mm y$^{-1}$. For
the Southern Interior basin, we found an apparent overestimation of c. 72 mm y$^{-1}$ (Table 1) which
suggests that at least some of this difference is realistic. Underestimation of precipitation may also go
some way towards explaining these differences: both regions are in transitional climates with a
relatively strong, non-orographic precipitation gradient of 900–1400 mm y$^{-1}$ (Yucatan) and 400–1100
mm y$^{-1}$ (Southern Interior), respectively. Combined with a low density of rainfall gauges (Hijmans et
al., 2005), these gradients make a systematic bias in rainfall estimates more plausible.




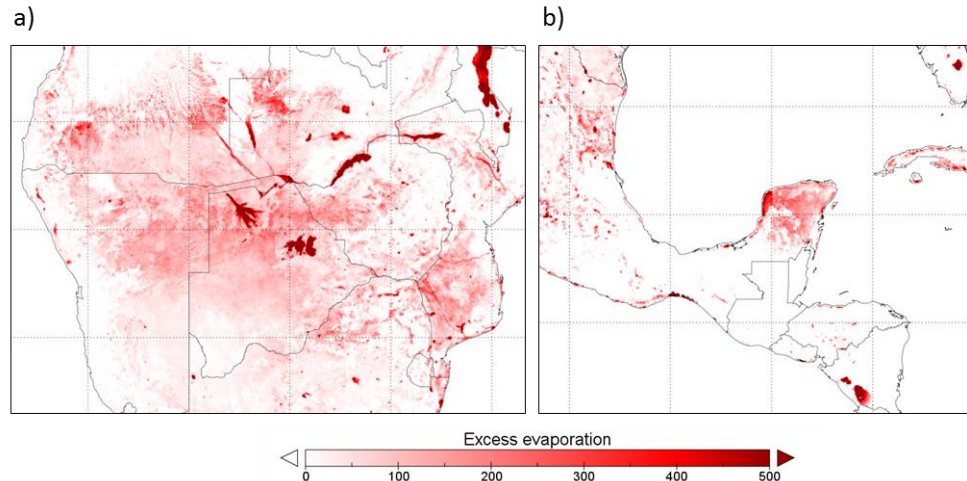


Figure 10. Mean difference between total evaporation and precipitation for 2001–2014 for (a) Botswana and (b)

the Yucatan peninsula, and surrounding areas.


*Uncertainty in irrigation water requirement estimation*

The total estimate of minimum irrigation water requirement ($I_0$) at the global scale was about a third

lower than previous model-based estimates (Siebert et al., 2010; Wada et al., 2014; Siebert and Döll,

2010). There are some likely explanations for this. Firstly, the diffuse distribution of areas equipped

for irrigation (Figure 2) means that the LST signal from irrigation will likely have been too small to

estimate the associated $I_0$ correctly everywhere. An insufficient LST signal is most likely for grid cells

and countries with a temperate and humid climate and highly distributed irrigation, such as the US,

where our estimate of $I_0$ was twice smaller than published previously. Conversely, irrigation

evaporation estimates should be more accurate in hot, arid regions with large and concentrated

irrigation, such as Egypt's Nile Delta (Figure 1). The temporal pattern of the evaporative fraction for

this grid cell corresponds well with that of vegetation cover (Figure 1e) and assumes values that

appear realistic, even more so when considering that only around 80% of the grid cell was irrigated

(Figure 1a).

Second, previous studies have estimated crop water use (and from that, $I_0$) using the FAO method of

Allen et al. (1998). This method assumes a well-growing crop not affected by ineffective or

insufficient irrigation, unfavourable weather, nutrition or soil, pests and diseases, or other growth-

limiting factors. The resulting crop water use estimates are likely to represent idealised conditions and

may be higher than actual water use.

Third, errors in irrigation area mapping are also likely to have played a role. It is noteworthy that the

MIRCA2000 mapping used here (Portmann et al., 2010) indicated that 100% of the grid cell in Figure





1a was equipped for irrigation. This is not the case: most unirrigated areas are settlements. Previous
studies will have assumed the entire area was available for irrigation and this difference alone would
cause their $I_0$ estimates for this particular grid cell to be 25% higher. While these numbers relate to
just a single grid cell, it serves to demonstrate that incorrect mapping of irrigation areas can have
considerable impact on our $I_0$ estimates. As another example, any irrigation outside the grid cells
indicated to have at least some irrigable area in the MIRCA2000 mapping would be wholly attributed
to non-irrigation forms of secondary evaporation.
Despite these caveats, it is highly likely that true irrigation water application is greater than our
estimate $I_0$, as it was defined as a hypothetical quantity that might occur under conditions of optimally
efficient irrigation. Previous studies have made similar assumptions. In reality, field-level irrigation
efficiency is reduced by additional drainage below the root zone and any surface runoff that may
occur. Further uncertainties are introduced through the necessary assumptions about rooting depth and
root zone storage capacity. The comparison with FAO-reported $W$ estimates suggests project
efficiencies that are lower than those assumed in previous studies, but the overall correlation between
country $I_0$ and $W$ volumes was high, and could not solely be attributed to differences in irrigated area
(Figure 5). A comparison of country $I_0$ and $W$ expressed as area-average rates indicates contrasts in
project efficiency that are expected in several cases. In other cases, values are outside a plausible
range. At least some of these poor estimates are likely related to the mentioned inaccuracies in
irrigation mapping (e.g., Chile and the United Arab Emirates in Figure 5b).
Overall, the method developed here shows a promising approach to estimate irrigation water use.
Estimation at an even higher spatial resolution should help to detect the LST signal more accurately
where irrigation areas are dispersed and so produce better estimates of $E'$. This provides a powerful
argument in support of 'hyper-resolution' water balance observation and modelling (Wood et al.,
2011). All satellite-derived inputs are available at a resolution that is about an order of magnitude
finer (500–1000 m) than used here, and computationally data assimilation at this resolution is also
already feasible. The main impediment is the resolution and quality of irrigation area mapping, which
is required to attribute secondary evaporation to irrigation and other sources. The $E'$ estimates
themselves may assist in mapping, along with information on temporal vegetation patterns, open
water mapping and relief, among others. This is an avenue we hope to pursue in future.
*Importance of secondary evaporation in the global water cycle*
Our analysis suggests that secondary evaporation makes a meaningful contribution to global
evaporation (8.1%) and reduces the amount of discharge to the oceans by c. 16%. At the global scale,
irrigation is responsible for only a small fraction of this reduction (c. 5%), with the remainder
occurring from water bodies and wetlands. These global averages hide significant regional variation.
For example, irrigation plays an important role in the evaporation of river flows in the Nile, Indus and
Murray-Darling basins, where most of the discharge is evaporated before reaching the ocean. About




half of total global secondary evaporation is from permanent freshwater bodies, including from some
very large water bodies such as the Caspian Sea, the Great Lakes, and the African Rift Valley Lakes.
We estimated global terrestrial evaporation to be 508 mm y$^{-1}$ per unit land area or 75.5·10$^{12}$ m$^3$ y$^{-1}$
total for 2001–2014, made up of 467 mm y$^{-1}$ or 69.6·10$^{12}$ m$^3$ y$^{-1}$ primary evaporation and 41.2 mm y$^{-1}$
or 6.1·10$^{12}$ km$^3$ y$^{-1}$ secondary evaporation. This is close to estimates derived from previous studies.
For example, Miralles et al. (2016) reported 13 estimates of terrestrial E, derived from a variable
combination of satellite observations and modelling, with an average value of 69.2·10$^{12}$ km$^3$ y$^{-1}$ and
coefficient of variation (CV) of ±10%. Schellekens et al. (2017) reported a mean of 74.5·10$^{12}$ km$^3$ y$^{-1}$
(CV of ±6%) for an ensemble of 10 state-of-the-art global hydrological models and land surface
models. Some of these differences are attributable to the differences in total area and period
considered, but the different datasets also includes secondary evaporation losses to different degrees.
Given these represent 8% of total evaporation, such inconsistencies help to explain differences
between estimates.
The partitioning between primary evaporation components is within the range of recently published
estimates, though noting that those ranges are broad (Table 4). Secondary evaporation is fully
responsible for open water evaporation and has no impact on wet canopy evaporation; both are a
logical consequence of the way these terms are conceptualised. It is estimated that global transpiration
and soil evaporation are both enhanced by about 4.5% due to secondary evaporation of surface and
groundwater resources. Irrigation is responsible for a tenth of this increase, with the remainder due to
natural processes. Because of the coupling between transpiration and carbon uptake, it can be
assumed that these enhancements will increase global carbon uptake by a similar proportion. Once
again these small contributions apply at global scale, but there are strong differences locally and
regionally.

Table 4. Estimated percentage of total (or, between brackets, primary) terrestrial evaporation (*E*)
contributed by different pathways, compared with estimates from two recent studies.

| Percent of total E | this study | Zhang et al. (2016) | Miralles et al. (2016) |
|---|---|---|---|
| wet canopy E | 16 (17) | 10 | 10-24 |
| transpiration | 57 (60) | 65 | 24-76 |
| soil E | 21 (23) | 25 | 14-52 |
| open water E | 4 (0) | − | − |


Thiery et al. (2017) simulated the global impact of irrigation using coupled land surface and
atmosphere models. They estimated an evaporation increase from irrigation of 418 km$^3$ y$^{-1}$; of similar
magnitude to the 300 km$^3$ y$^{-1}$ we found. Despite this small contribution to total global evaporation,
their modelling did predict small but meaningful reductions in high-temperature extremes over and
near large irrigation areas; irrigation rates tend to be highest during hot and dry conditions. To the best





of our knowledge, there have been no studies on the impact of wetlands and water bodies on regional

and global climate so far. Given that we estimate these other forms of secondary evaporation to be

twenty times greater than from irrigation, their impact on the atmosphere should be significant.

**Conclusions**

We presented a methodology to assimilate thermal satellite observations into a global hydrological

model W3 at a resolution of 0.05° to estimate secondary evaporation of surface and groundwater

resources. In addition, we used a simple irrigation water balance model to estimate minimum

irrigation requirement ($I_0$) globally. Our main conclusions are as follows.

(1) The method developed produces realistic temporal and spatial patterns in secondary evaporation.

Accounting for secondary evaporation measurably improved water balance estimates for large closed

and open basins, reducing bias in the overall water balance closure from +38 to +2 mm y$^{-1}$.

(2) Our $I_0$ estimates were lower than country-level estimates of irrigation water use produced by other

model estimation methods, for three reasons. Firstly, at the 0.05° resolution, much of global irrigated

land occupies only a small part of individual grid cells and may not reduce LST sufficiently to be

accurately estimated. Second, our $I_0$ estimates reflect actual evaporation, which can be lower than

idealised crop water use estimates used in previous studies. Third, spatial errors in irrigation area

mapping directly affect the attribution of secondary evaporation to irrigation. Overall, actual irrigation

application will most likely be higher than estimated here but possibly lower than reported previously.

(3) The role of irrigation water use in secondary evaporation is minor at the global scale, accounting

for 5% of total secondary evaporation and 0.4% of total terrestrial evaporation. Nonetheless, water

withdrawals and irrigation evaporation are an important part of the water balance in some regions.

(4) Around 16% of globally generated water resources evaporate before reaching the oceans,

enhancing total terrestrial evaporation by 8.8%. Of this secondary evaporation, 5% is evaporated from

irrigation areas, 58% from water bodies, and 37% from other surfaces.

(5) Lateral inflows of surface and water resources were estimated to increase global plant

transpiration by c. 4.5%. The impact on global carbon uptake would be expected to be of similar

magnitude. Previous studies have predicted that irrigation evaporation affects regional and global

climate. Given evaporation from wetlands and permanent water bodies is an order of magnitude

larger, their impact on the climate system should be pronounced.

There is scope for further improvement in accounting for natural and anthropogenic secondary losses

by applying the model-data assimilation approach developed here at higher resolution. This is

conceptually straightforward and computationally achievable. Key developments required include





more accurate and detailed dynamic observational data on surface water dynamics and more accurate
mapping of areas equipped for irrigation.
**Data availability**
The 5-km water balance estimates presented here will be available via *http://www.wenfo.org/wald/*.

**Acknowledgements**
The MODIS products were retrieved from the online Data Pool, courtesy of the NASA EOSDIS Land
Processes Distributed Active Archive Center (LP DAAC), USGS/Earth Resources Observation and
Science (EROS) Center, Sioux Falls, South Dakota, https://lpdaac.usgs.gov/data_access/data_pool.
Albert van Dijk was supported under Australian Research Council's Discovery Projects funding
scheme (project DP140103679).

**Author contribution**
AVD conceptualised the study. JS, HB, AW and GD developed global input data for the modelling.
MY developed the remote sensing evaporation scheme. LR assisted in the development of the data
assimilation approach. AVD carried out the analysis and wrote the first draft manuscript. All other
authors contributed to the analysis, interpretation and writing.

**Appendix A. Global data sets used**
•    *Climate* forcing data used included the MSWEP multi-source merged precipitation product

version 1.1 (Beck et al., 2017) and the WFDEI dataset version 1 (Weedon et al., 2014) for other

atmospheric variables (short- and longwave down-welling radiation, screen-level air temperature

and humidity, wind speed, snowfall fraction, and surface pressure). Air temperature and

precipitation were downscaled to 0.05° using the HYDROCLIM long-term monthly climatologies

of air temperature and precipitation (Hijmans et al., 2005).

•    *Terrain properties* used include slope and points of the per-cell distribution of height above

nearest drainage (HAND) that were derived by the authors from the global SRTM Digital

Elevation Model (DEM) combined with the GTOPO30 DEM beyond 60° latitude. Flow direction

was derived from the HydroSheds dataset (Lehner et al., 2008) extended with the hydro1k product

beyond 60°.

•    *Surface and vegetation properties* were largely derived from ESA's GlobCover version 2.2

mapping product, based on 300m resolution observations from the optical MERIS instrument

between December 2004 and June 2006 (Bicheron et al., 2008). From these data, we derived



0.05° grids representing fractions of permanent water, ice and artificial surfaces, as well as
fractions of deep- and shallow-rooted vegetation estimated from the land cover fractions.
Vegetation height estimates were those derived from ICESat-GLAS measurements by Simard et
al. (2011)
• *Snow* model parameters for the conceptual HBV were derived by Beck et al. (2016).
• *Soil properties* were derived from the GSDE dataset (Shangguan et al., 2014), a global gridded
data set of soil properties. Gridded soil parameters that were derived include saturated
conductivity, saturated water content, bubbling pressure and the pore size index lambda
(following Brooks and Corey, 1964).
• *Aquifer properties* used include gridded estimates of shallow aquifer porosity from the
GLHYMPS data set (Gleeson et al., 2014), whereas gridded estimates of groundwater recession
constants were obtained from the GSDC dataset (Beck et al., 2015).
All satellite products assimilated in model run time were ultimately derived from NASA's MODIS
instruments on the Aqua and Terra satellites.
• *Surface albedo and reflectance data* were derived from the combined MODIS Terra/Aqua 8-day
composite products resampled to 0.05° resolution global grids (MCD43). White-sky albedo was
derived from the MCD43C3.005 product, whereas percent snow cover and nadir reflectances in
the red (Band 1), near infrared (Band 2), blue (Band 3) and shortwave infrared (Band 6) were
obtained from the MCD43C4.005 product.
• *For leaf area index*, the MODIS GLASS product (Xiao et al., 2014) resampled to 0.05° resolution
was used.
• *Land surface temperature* was derived from the MOD11C1.006 product (Wan, 2015), providing
daily estimates of land surface temperature based on MODIS Terra observations resampled to
daily grids. Only optimum quality data were used, indicated by a daily quality control index value
of zero. Except the GLASS LAI product (downloaded from http://glcf.umd.edu/data/lai/) all
satellite data were downloaded through the NASA data portal.

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
