# Peer review of "Global 5-km resolution estimates of secondary evaporation including irrigation through satellite data assimilation"

_Hydrology and Earth System Sciences, 2017_

## Referee Comment (RC1) · Anonymous Referee #1 · 22 Mar 2018

I really enjoyed reading this paper and find that it has valuable contributions towards a global irrigation demand estimate. It provides an interesting new perspective on obtaining global irrigation estimates. However, while I was reading the manuscript I did have several questions and concerns. Please find below the major and minor comments I have after reading the manuscript.

Major comments: I find that the introduction is somewhat incorrect and building up to something that is not really happening in the objective of the paper. The authors mention how all kind of modelling efforts will not produce independent and accurate estimates of irrigation water demand, but after the reading the objective I can only

conclude that they will themselves do modelling as well. I think it would be good to refocus the introduction and state that models can have a valuable contribution but have their limitations and highlight how the authors would like to resolve these limitations.

Line 123-139 In the introduction it is stated that most models do not include lateral water flow, but then again, the others already give some model example where lateral flow is included. Arguably the lateral flow in the routing is not included, but that is also the case for the model used in this study. I do not fully understand why this model is different from the other global water balance models out there. They authors should do a better job to highlight this, to emphasize why this model is better suited for this excursive than others

Line 125 -126 The quality of the forcing data is really low, how do the authors think this will impact the simulations and consequently the evaporation estimates?

Line 184-185 and Figure 1, I have the strong feeling that the model is biased in its estimates of E. Therefore, this would violate the basic assumption of a normal distribution with a mean of 0 around the observations. In addition, the authors cutoff the E' updates, which is in my opinion another violation of the EnKF. I feel the others should make sure that the model is bias free before implementing a DA technique like the EnKF. Otherwise, they can show the global biases to convince the reader that this is only the case for Figure 1, but I have to strong suspicion that it is also a problem for other regions (as for most models). I think the authors should address this large limitation in their discussion or somewhere else in the manuscript.

The manuscript could significantly benefit from a flowchart describing the full updating, calibration, nudging and assimilation procedure. Which variable are subject to what and where and how? The manuscript is difficult to follow without.

Line 253-254 Why is the increase in the estimation evaporation not from missing model processes? Incorrect vegetation parameterization or something else. This assumption is vital for the manuscript and is not really supported by argument on the model's quality

to estimate evaporation in general. Has the model been validated against independent evaporation estimates?

In addition, to the previous comment, the authors have not mention other forms of water use. I see no inclusion of domestic or industrial water use in the model nor in the estimates? Maybe these abstractions cause the errors in water basin closures. Can the authors exclude this and/or provide estimates of these quantities for the river basins? This would inform the reader if irrigation evaporation has a real significant contribution or if it is simply correcting for the fact that other sources of water use are missing in the model

Minor comments: Line 129-134 are the calibrated parameter spatially consistent or are they really tuned to the individual basins?

Line 134-135 Does the model have any lateral flow simulations of groundwater or surface water?

Line 150 a nudging factor of 0.99 is rather high, does this mean that the model is almost always wrong?

Line 156-159 what is the spatial resolution of the Tair forcing, since it is very important for the LST simulations

Line 177 15degree, does this mean that the LST is spatially average over a 1500 by 1500km area???

Line 508-510 the true error can also be larger... It is not said that it will be smaller due to the representativeness error.

Line 581-583 As far as I understand most other models use sub-grid parameterization, which would allow for a partial coverage of the grid cell by irrigation areas. This statement is therefore potentially incorrect and should be removed to avoid misinforming the reader

Line 619-623 I feel the units are incorrect, I guess the first estimates should be 75.5*10ˆ12 Km3 y-1 (as well as for the other estimates from this study, which are now 1000 times lower than other studies)

---

## Referee Comment (RC2) · Anonymous Referee #2 · 11 Apr 2018

This study, "Global 5-km resolution estimates of secondary evaporation including irrigation through satellite data assimilation, " presents model-based estimate of the ET from secondary sources. The data produced (provided) is potentially useful, and the modeling framework is commendable for using multiple constraints and using satellite remote sensing in model-data fusion. The analyses presented in the manuscript are scientifically sound, but I have some comments or suggestions that would, hopefully, be useful for the authors to improve the analysis manuscript.

General: First of all, I find the manuscript a bit unbalanced in terms of contents. There is a lot of focus on methods and equations (esp. for irrigation), but relatively a few

figures for results. This makes the manuscript very tedious to read with a lot of text and information. At the same time, some information that are critical to assess the results are either missing or in the appendix. For example, forcings and their spatial disaggregation, model formulations of LE and H, etc.

Definition of the secondary evaporation: There is no description on how groundwater's contribution to LE/ET is a secondary source. In an idealistic theoretical situation, the capillary flux from groundwater will replenish soil moisture (at some point when the soil moisture is drying up), which would eventually increase LE. It is not clear if the model considers such capillary flux processes explicitly. I am curious about what fraction of 'other' sources is actually coming from groundwater-soil-LE pathway, and not groundwater-baseflow-surface water-LE pathway. The first one may have a critical influence on vegetation and carbon cycle processes.

Assimilation of LST into model: In the assimilation of LST into model, the basic assumption is that the model-simulated partitioning of the energy fluxes (H and E) are correct. The corrections or 'nudges' for LST are back-calculated from the modelled H, and these are propagated through spatial patterns of observed LST. But, there is no explanation of how 'background' H and LE are calculated in the model. Perhaps, these may be inferred from previous papers/reports on the model (?), but they are so critical for this study and results presented herein, they deserve to be in this manuscript. One information that is imperative is whether the parameters of the modelled LE and H were optimized or not. If not, are the used parameter values are reasonable for a global-scale application?

Related to the above point, validation for model simulated LE and H is not shown or discussed. There are references to a previous study or an unpublished work but the findings of this study also warrant a section on evaluations at the global scale. I am aware that observed global ET and H data are not available, but a comparison with either FLUXNET observations (for sites) or other satellite-based ET products can provide a valuable benchmark.

Estimation of irrigation water use: Assumption of rooting depth: The parameter smax is dependent on the assumed rooting depth. The manuscript would benefit from a discussion on how these parameters vary globally, and to what extent do this variation affects the estimation of secondary evaporation from irrigated area.

Evaluation against discharge observations: In my subjective judgment, the improvement in the basins with discharge < 300 mm/y is mostly driven by Paraná because it has discharge with the largest magnitude. In reality, the river basins with large irrigation water withdrawal/use are also equipped with dams and are not of run-of-river type (with no reservoir). The secondary evaporation from these 'dammed' rivers also comprise of evaporation from reservoirs. So, in my opinion, it would be helpful to include the information of reservoir volume (e.g., from GranD database) in the analysis or the figure. This is important because the water evaporated from the reservoirs might actually be significant, especially because the irrigation requirement/use from this study is much lower than previous estimates.

Comparison with previous estimates: The manuscript addresses the minimum irrigation water requirement, which I understood as the actual gross irrigation water use (gross because it has both bare soil evaporation in irrigated areas+transpiration by crops). In most previous modeling studies, difference between PET and ET is used to calculate irrigation water requirement (and withdrawal). Current manuscript rightly points that there are several limitation to ET from irrigated areas. Despite that, it would make sense to compare the difference between PET and ET (Priestley Taylor is already used in the current study) with the bias of I0 against withdrawal.

Forcing variables: The results of this study are extremely dependent on the biases in the WFD forcing data as well as the spatial patterns of HYDROCLIM data. It is not clear from the current analysis if the biases in secondary evaporation are related to WFD magnitude (over a half degree grid) or the spatial patterns of HYDROCLIM (over 0.05 deg grids).

Temporal variation of secondary evaporation: I would have really learnt a lot on what is driving the secondary evaporation if there was a discussion on temporal variation of secondary evaporation at the global scale. This would provide insights on whether the secondary evaporation increases in wet season (for e.g., in water bodies such as wetlands and river channels because the surface area becomes larger) or in dry season in which the groundwater access by plant can be expected to be maximum.

Evaporation larger than precipitation in southern Africa and Yucatan: The discussion focuses on the biases in the precipitation. If total E (primary + secondary) were correct, the signal should appear in the water storage changes. In that case, GRACE satellite measurements should show a declining terrestrial water storage. A comparison on loss of storage in the study period and the total E – P would provide a great motivation for future studies on what are driving such changes. Essentially, this would already help in refining the potential causes of the negative water budget.

Editorial Comments:

Line 1: In my opinion, 'estimates' should be replaced by 'simulations'. Essentially, the results are dependent on hydrological model simulations.

Line 232: i=1,26 can be replaced by just 26.

Line 259: There is no description for what Pg is. I assumed that it is precipitation for the grid cell.

Figures 6-9: I recommend using the same color maps and scales in these figures. It is a bit confusing because the same color 'blue' means a different value in different figures.

Table 1: Just curious that observed discharge in Nile is 0. Fascinating that no water from such large river basin reaches the ocean.

Line 534: can have affected –> can affect or could have affected
Line 621: wrong units: km3/yr –> m3/yr

Line 671-673: –> Before reaching the ocean is misleading because a fraction of the open water evaporation is from rivers which do not drain to ocean (e.g., inland lakes).

Line 674-678: Does the groundwater include baseflow-river-ET and groundwater capillary flux-soil moisture-ET? I am not sure if the second process can be categorized as the secondary evaporation.

---

## Author Comment (AC1) · 3 May 2018

We thank the reviewer for their comments and are glad that they enjoyed the m/s. Below we address the issues raised.

"The authors mention how all kind of modelling efforts will not produce independent and accurate estimates of irrigation water demand, but after the reading the objective I can only conclude that they will themselves do modelling as well. I think it would be good to refocus the introduction and state that models can have a valuable contribution but have their limitations and highlight how the authors would like to resolve these limitations."

[Figure]

It is certainly true that our approach requires a model to assimilate the satellite observations into, and it is also true that additional assumptions are needed to translate water use estimates to irrigation water demand. We meant to emphasise that our method differs from existing methods in that it does not require mapping of the area irrigated or the extent of wetlands to then explicitly simulate secondary evaporation. Instead it is inferred from the observations. In revising, we will look for some textual changes to avoid the wrong impression.

"I do not fully understand why this model is different from the other global water balance models out there. They authors should do a better job to highlight this, to emphasize why this model is better suited for this excursive than others"

The overall approach is different from existing models in that secondary evaporation is constrained by the satellite observations, rather than the result of simulation. If the reviewer refers to the W3RA model used in assimilation, then we believe a similar approach could be applied to other models, provided they have a coupled water and energy balance model and provided they are extended for data assimilation in the way described.

"The quality of the forcing data is really low, how do the authors think this will impact the simulations and consequently the evaporation estimates?"

We do not think that all forcing data can reasonably be considered of really low quality in any objective sense, but some of the inputs create greater uncertainty than others. The model takes several different forcing data as input, the evaporation estimates are not equally sensitive to all of them, and the quality of forcing data also varies spatially. Hence it is impossible to give a quantitative answer to the question. With regard to evaporation specifically, one observed issue is that of heterogeneous biases in air temperature in regions with strong relief. Fortunately, secondary losses occur mostly in areas with low relief (see l. 162-167).

"Line 184-185 and Figure 1, I have the strong feeling that the model is biased in its

estimates of E. Therefore, this would violate the basic assumption of a normal distribution with a mean of 0 around the observations. In addition, the authors cut-off the E' updates, which is in my opinion another violation of the EnKF. I feel the others should make sure that the model is bias-free before implementing a DA technique like the EnKF. Otherwise, they can show the global biases to convince the reader that this is only the case for Figure 1, but I have to strong suspicion that it is also a problem for other regions (as for most models). I think the authors should address this large limitation in their discussion or somewhere else in the manuscript."

Does the reviewer mean the model is biased in the absence of secondary evaporation (i.e. in drylands)? We do not have evidence for that. We have evaluated the ("offline" or background) model against evaporation rates reported by the global Fluxnet network for non-irrigated environments and did not find a bias (l. 505-510). The reviewer suggests showing global biases, but there are no global ET observations (other than Fluxnet) to calculate those from. However, we do compare with estimates from other models and find that ours are well within that range (l. 619-629). We did not use EnKF but nudging based on energy balance model inversion. The reviewer is correct that we did cut off the E' updates. It was necessary to maintain internal physical consistency, but it is true that it may have introduced bias, particularly if real E was consistently higher than the available energy, for example, due to biases in meteorological forcing data. We can mention this in revising the m/s.

"The manuscript could significantly benefit from a flowchart describing the full updating, calibration, nudging and assimilation procedure. Which variable are subject to what and where and how? The manuscript is difficult to follow without."

Thank you for the suggestion. Unfortunately, data assimilation procedures can be difficult to explain and tedious and confusing to read. We will add a flow diagram to attempt to illustrate the procedure better.

"Line 253-254 Why is the increase in the estimation evaporation not from missing model
processes? Incorrect vegetation parameterization or something else. This assumption is vital for the manuscript and is not really supported by argument on the model's quality to estimate evaporation in general. Has the model been validated against independent evaporation estimates?"

This is discussed in l. 562-587. The assimilation of satellite vegetation observations goes some way to address errors in vegetation parameterization. However, the (necessary) assumption that the assimilation increment is due to irrigation has uncertainty associated with if - but only if - a large part of the grid cell is occupied by non-irrigated land. Hence also the recommendation that our approach should work better at higher resolution, if good irrigation area mapping was available. We hope to pursue this line of enquiry (see l. 600-609). Regards validation: see previous comment.

"In addition, to the previous comment, the authors have not mention other forms of water use. I see no inclusion of domestic or industrial water use in the model nor in the estimates? Maybe these abstractions cause the errors in water basin closures."

Domestic and industrial water use are not considered because these are typically non-consumptive uses (i.e., the water is returned to the environment after use). Possibly the main exception to this would be irrigation in urban landscapes, which the irrigation mapping does not capture well or at all. If those uses lead to surface cooling then the LST data assimilation will still have increased E estimates and so they are implicitly accounted for. In practice, consumptive urban or industrial water uses are unlikely to have a meaningful impact on the water balance of large basins.

"Line 129-134 are the calibrated parameter spatially consistent or are they really tuned to the individual basins?"

Neither, they vary spatially as a function of climate aridity and land cover using predictive relationships derived from model calibration to evaporation, soil moisture and streamflow from a very large number of sites and small and unregulated catchments, respectively.

[Figure]

"Line 134-135 Does the model have any lateral flow simulations of groundwater or surface water?"

No, only grid-based routing (l. 135)

"Line 150 a nudging factor of 0.99 is rather high, does this mean that the model is almost always wrong?"

Poor at predicting highly dynamic surface water extent, you could say, yes. Like all global models, we believe.

"Line 156-159 what is the spatial resolution of the Tair forcing, since it is very important for the LST simulations"

$0.05°$ using HYDROCLIM to downscale of the WFDEI data (l. 701-706). We agree that correct Tair is important.

"Line 177 15degree, does this mean that the LST is spatially average over a 1500 by 1500km area???"

Correct, but note that this does not imply that LST is assumed homogenous across the area, this calculation is to remove the mean bias between daytime LST and time-of-overpass LST.

"Line 508-510 the true error can also be larger. . . It is not said that it will be smaller due to the representativeness error."

At least in theory, yes. Agreed.

"Line 581-583 As far as I understand most other models use sub-grid parameterization, which would allow for a partial coverage of the grid cell by irrigation areas. This statement is therefore potentially incorrect and should be removed to avoid misinforming the reader"

Disagree. The MIRCA2000 mapping suggests the grid cell is 100% equipped for irrigation. To our knowledge, the published models assume that the entire equipped area is irrigated so the statement holds. Of course, that assumption could be changed in principle, but that would simply move the problem.

"Line 619-623 I feel the units are incorrect, I guess the first estimates should be 75.5*10Ë Ę12 Km3 y-1 (as well as for the other estimates from this study, which are now 1000 times lower than other studies)

We believe the units are correct. We could have written 75.5*10ˆ12 m3 y-1 as 75,500 km3 y-1 but felt using base units (m) was more appropriate.

---

## Author Comment (AC2) · 5 May 2018

We thank the reviewer for their positive and constructive comments. Below pls find a response to the issues raised.

**"First of all, I find the manuscript a bit unbalanced in terms of contents. There is a lot of focus on methods and equations (esp. for irrigation), but relatively a few figures for results. This makes the manuscript very tedious to read with a lot of text and information. At the same time, some information that are critical to assess the results are either missing or in the appendix. For example, forcings and their spatial disaggregation, model formulations of LE and H, etc."**

[Figure]

We are sorry the m/s was tedious to read. We accept that the technical detail of the modelling and data assimilation can be a bit tedious, which is why we tried to minimise that aspect in the main text by transferring some of the material to the appendix and referring to previous publications where possible. In principle, we could 'spruce up' the m/s with additional figures, but there are in fact already 10 figures, several of them multi-panel ones, and hence we are not sure it would make the m/s less tedious to read. We will include a new figure illustrating the workflow, as requested by the other referee. At the same time, however, the referee also asks for additional material to be included that would likely make the m/s even more tedious for readers not overly interested in the modelling details. Given the model theory and formulation is already available elsewhere we hesitate to overburden this paper with it. The energy balance equation is the main model component of relevance here, which is why we included that aspect of the theory. We look for guidance from the editor as to what changes should be made to make the m/s less tedious yet provide any additional information deemed critical to assess the results. Possible options could include putting detail into the supplementary material, for example.

**"Definition of the secondary evaporation: There is no description on how groundwater's contribution to LE/ET is a secondary source. In an idealistic theoretical situation, the capillary flux from groundwater will replenish soil moisture (at some point when the soil moisture is drying up), which would eventually increase LE. It is not clear if the model considers such capillary flux processes explicitly. I am curious about what fraction of 'other' sources is actually coming from groundwater-soil-LE pathway, and not groundwater-baseflow-surface water-LE pathway. The first one may have a critical influence on vegetation and carbon cycle processes."**

The model does consider capillary fluxes, but in the offline model those are ultimately constrained by longer-term local rainfall, and therefore do not constitute secondary evaporation (i.e., it is derived from locally recharged, unconfined groundwater rather than lateral groundwater inflows). As our study demonstrates, data assimilation helps

to estimate secondary evaporation from non-local water sources, but does not directly attribute it to a water source – that requires ancillary data. In some cases, the secondary evaporation may be from irrigation with water pumped from confined aquifers (which bypasses the capillary rise pathway). In other cases, it is possible that secondary evaporation is inferred, e.g. because rainfall is underestimated capillary rise or deep root water uptake is more important than predicted by the background model (e.g., because the vegetation is more deeply rooted or groundwater is closer to the surface than assumed). There is obviously much more to be done to understand the global water balance in full detail. Our data provide a means of prioritising regions where there appears to be hydrological behaviour that is not easily explained by the background model, and therefore is worthy of further investigation.

**"Assimilation of LST into model: In the assimilation of LST into model, the basic assumption is that the model-simulated partitioning of the energy fluxes (H and E) are correct. The corrections or 'nudges' for LST are back-calculated from the modelled H, and these are propagated through spatial patterns of observed LST. But, there is no explanation of how 'background' H and LE are calculated in the model. Perhaps, these may be inferred from previous papers/reports on the model (?), but they are so critical for this study and results presented herein, they deserve to be in this manuscript."**

The basic assumption is actually not that the partitioning of H and LE is correct, but rather, that the estimated total available energy (A=H+LE) is correct. Data assimilation may change the estimate of H and through that LE=A-H. The background H and LE are estimated using the conventional Penman-Monteith approach (l. 186). However, we agree that we did not explain this very well and also did not provide much detail on the way PM theory was implemented and parameterised. Perhaps this is also the important detail the referee referred to in the opening comment. This information is indeed detailed in the model documentation, but we agree that that is an important aspect and will include more details in revising the m/s. It is a bit tedious due to the consideration of several evaporation pathways, and hence we might include it as an

appendix.

**"One information that is imperative is whether the parameters of the modelled LE and H were optimized or not. If not, are the used parameter values are reasonable for a global-scale application?"**

They were not optimised. The most important parameter overall, surface conductance, was predicted from satellite-observed surface reflectances following Yebra et al. (2013) and tuned using a large database of evaporation measurements (FLUXNET). Another important parameter, vegetation height (affecting aerodynamic conductance) was derived from Lidar remote sensing by Simard et al. (2011).

**"Related to the above point, validation for model simulated LE and H is not shown or discussed. There are references to a previous study or an unpublished work but the findings of this study also warrant a section on evaluations at the global scale. I am aware that observed global ET and H data are not available, but a comparison with either FLUXNET observations (for sites) or other satellite-based ET products can provide a valuable benchmark."**

We have performed this analysis and mention the results in the text (l. 507-508). In revising the m/s we could include a figure with those results if the editor feels it adds value. We did not do so as it could mistakenly be interpreted as a validation of the data assimilation procedure, which it is not: the vast majority of flux towers are in environments without secondary evaporation.

**"Estimation of irrigation water use: Assumption of rooting depth: The parameter smax is dependent on the assumed rooting depth. The manuscript would benefit from a discussion on how these parameters vary globally, and to what extent do this variation affects the estimation of secondary evaporation from irrigated area."**

This is explained in l. 232-236. Essentially, we follow the published methodology of Siebert and Döll (2010). The assumptions made here, in fact, do not affect the estimation of secondary evaporation at all. What it does affect is the calculated irrigation efficiency and therefore the estimate of irrigation water use. This is a perhaps subtle, but important distinction.

**"Evaluation against discharge observations: In my subjective judgment, the improvement in the basins with discharge < 300 mm/y is mostly driven by Paraná because it has discharge with the largest magnitude. In reality, the river basins with large irrigation water withdrawal/use are also equipped with dams and are not of run-of-river type (with no reservoir). The secondary evaporation from these 'dammed' rivers also comprise of evaporation from reservoirs. So, in my opinion, it would be helpful to include the information of reservoir volume (e.g., from GranD database) in the analysis or the figure. This is important because the water evaporated from the reservoirs might actually be significant, especially because the irrigation requirement/use from this study is much lower than previous estimates."**

Actually, it is largely also due to the improved water budget for closed basins (dots on the vertical axis) and several other basins (e.g., Indus). Our methodology does use remotely sensed water extent, and that would include reservoir surface area, so evaporation from reservoir surfaces would be included in the estimates.

**"Comparison with previous estimates: The manuscript addresses the minimum irrigation water requirement, which I understood as the actual gross irrigation water use (gross because it has both bare soil evaporation in irrigated areas+transpiration by crops). In most previous modeling studies, difference between PET and ET is used to calculate irrigation water requirement (and withdrawal). Current manuscript rightly points that there are several limitation to ET from irrigated areas. Despite that, it would make sense to compare the difference between PET and ET (Priestley Taylor is already used in the current study) with the bias of I0 against withdrawal."**

Unfortunately, we did not fully understand the analysis the referee proposes. We do compare I0 to withdrawal in Fig 5 and l. 435-449, and this does provide some useful

insights, discussed in l. 561-599. We could compare irrigation area ET to PET (as done for example in Fig. 1e) but are not sure how to summarise such a comparison globally or what it would demonstrate.

**"Forcing variables: The results of this study are extremely dependent on the biases in the WFD forcing data as well as the spatial patterns of HYDROCLIM data. It is not clear from the current analysis if the biases in secondary evaporation are related to WFD magnitude (over a half degree grid) or the spatial patterns of HYDROCLIM (over 0.05 deg grids)."**

The term 'extremely' is subjective, but given the Penman-Monteith energy balance approach used, the evaporation estimates will depend on the meteorological forcing data, as does any method to estimate evaporation. We used the relative spatial patterns in HYDROCLIM only to adjust air temperature. Because we only assimilated satellite LST in areas with modest relief, we do not expect that the downscaling will have had much effect on secondary evaporation estimates. We also suspect that biases in air temperature in the WFD forcing data may, in fact, be less important than uncertainties in the radiation balance, wind speed, and perhaps specific humidity.

**"Temporal variation of secondary evaporation: I would have really learnt a lot on what is driving the secondary evaporation if there was a discussion on temporal variation of secondary evaporation at the global scale. This would provide insights on whether the secondary evaporation increases in wet season (for e.g., in water bodies such as wetlands and river channels because the surface area becomes larger) or in dry season in which the groundwater access by plant can be expected to be maximum."**

We thank the reviewer for this interesting suggestion and will seek to add some analysis around this. Based on our results so far, we suspect that secondary evaporation will be greatest in the warm season due to the importance of evaporation from slowly changing water bodies. However, we can certainly calculate these patterns and will look into it.

**"Evaporation larger than precipitation in southern Africa and Yucatan: The discussion**

focuses on the biases in the precipitation. If total E (primary + secondary) were correct, the signal should appear in the water storage changes. In that case, GRACE satellite measurements should show a declining terrestrial water storage. A comparison on loss of storage in the study period and the total E – P would provide a great motivation for future studies on what are driving such changes. Essentially, this would already help in refining the potential causes of the negative water budget."

Once again we thank the reviewer for the suggestion. Some knowledge of GRACE-based trends was on our mind in interpreting the results, but we did not make this explicit. Essentially, a previous GRACE model-data assimilation study some of the authors were involved in (Van Dijk et al., 2014: doi:10.5194/hess-18-2955-2014) inferred that water storage did decrease slightly over the Yucatan peninsula between 2003 and 2012 (slightly different from the 2001-2014 period in the present m/s), but increased quite strongly in southern Africa. Neither trend was predicted by an ensemble of hydrological models (particularly not the African case), which led us to suspect deficiencies in the rainfall estimates driving those models. We will expand the discussion along these lines.

Editorial Comments: # "Line 1: In my opinion, 'estimates' should be replaced by 'simulations'. Essentially, the results are dependent on hydrological model simulations."

We disagree; satellite observations were assimilated to make the results less dependent on model simulations.

**"Line 232: i=1,26 can be replaced by just 26."**

We used this notation to make the meaning of i in Ai in the same sentence clear.

**"Line 259: There is no description for what Pg is. I assumed that it is precipitation for the grid cell."**

The referee is right. Apologies, we will revise this.

**"Figures 6-9: I recommend using the same color maps and scales in these figures.**

It is a bit confusing because the same color 'blue' means a different value in different figures."

Thank you, we can change that.

**"Table 1: Just curious that observed discharge in Nile is 0. Fascinating that no water from such large river basin reaches the ocean."**

Agreed.

**"Line 534: can have affected –> can affect or could have affected"**

Agreed, thank you.

**"Line 621: wrong units: km3/yr –> m3/yr**

Agreed, thank you.

**"Line 671-673: –> Before reaching the ocean is misleading because a fraction of the open water evaporation is from rivers which do not drain to the ocean (e.g., inland lakes).**

We do not think this is misleading. Our phrasing was chosen for pragmatic reasons, although there is also a conceptual argument. The pragmatic reason was that, in identifying closed basins, we found it challenging to separate "truly" closed basins from basins that DEM analysis suggested were closed but which actually did appear to have an overflows according to independent reports. Surprisingly, it appears that there is no reliable global map of closed basins, and it took background research to identify the basins shown in Fig. 3. There were many other basins that the DEM suggested were closed but where we were not able to confirm that, meaning we ultimately did not identify all closed basins and therefore cannot make the distinction between secondary evaporation from (all) closed basins and all ocean reaching rivers. The conceptual reason is that the referee's argument can, in fact, be turned around: those rivers in 'closed basins' do not drain to the ocean because open water evaporation is so high.

The difference between closed and ocean-draining basins is a threshold (lake) level, and some basins currently switch between these states depending on the difference between rainfall and evaporation, many others did in the past. We do accept that there are closed basins that would require a very large increase in rainfall indeed (or decrease in evaporation) to top the overflow threshold and start draining to the ocean, but it does mean that there is no fundamental difference between 'closed' and 'open' basin. We do believe that a map of all (currently) closed basins would be a valuable information source for water balance studies and are currently looking into producing one using DEM data of higher accuracy and resolution, but early indications are that it requires intensive quality control. If it had existed, we would have made the distinction.

**"Line 674-678: Does the groundwater include baseflow-river-ET and groundwater capillary flux-soil moisture-ET? I am not sure if the second process can be categorized as the secondary evaporation."**

We are not entirely sure how to interpret this question. The primary evaporation estimates by the model do include the effect of capillary rise. However, if the primary evaporation estimates are too low data assimilation increases those estimates, and the difference will be (perhaps partly or wholly incorrectly) ascribed to secondary evaporation from lateral inflows. We discuss this in l. 501-504.

---

## Author Response (AR1)

"Global 5-km resolution estimates of secondary evaporation including irrigation through satellite data assimilation" by Albert I.J.M. van Dijk, Jaap Schellekens, Marta Yebra, Hylke E. Beck, Luigi J. Renzullo, Albrecht Weerts, and Gennadii Donchyts

**Response to Editor and Reviewers**

[1] We thank the editor and two reviewers for their comments and suggestions. We have made several revisions accordingly, which we explain below. The line numbers refer to the annotated m/s version with changes marked.

**Response to Editor**

I have read the very detailed comments from both reviewers, as well as the detailed responses you have provided to each of these comments. It strikes me that the points raised primarily require clarification, as they stem from confusion on the part of the reviewer.

You have made several clear comments on how you propose to update the manuscript to help clarify the points raised. Mostly your suggestions are clear. However, in several cases you leave it as reply to the comment. I would encourage you to consider adding one or two words/phrases to help clarify the confusion. This is useful for those readers who may be equally confused but are not inclined to go through all the interactive comments and replies (which I would suspect to be the majority).

[2] Agreed. We have made textual changes to pre-empt similar questions or issues wherever we saw an opportunity, as explained in our responses below.

**Response to Reviewer #1**

I think the comment on WR3A models is different should be addressed in the manuscript, as you are not clear if you will do this. It is I think important to point out that the structure of the model is important in being able to apply the approach presented, which precludes other perhaps simpler water balance type models.

[3] Agreed, we have added such a sentence (see response [11]).

On the flow diagram. This may indeed be useful, but as it primarily has the purpose to clarify, you may consider including it in the supplementary material.

[4] We have made and included such a flow diagram. In revising we initially included it in the supplement, but it seemed a bit out of place there and so we moved it into the main text. We would gladly take advice, however.

**Response to Reviewer #2**

I agree that it is not necessarily beneficial to the readability of the manuscript if all the details on model formulations are included in the main manuscript. In fact this may even be detrimental. The suggestion to include these in the supplementary material is I think a good one, but I would also restrict that to not overburden these, as references are indeed given to more complete descriptions. I trust the authors can assess what additional information is useful here. The authors should of course clearly refer the reader to the supplementary material as appropriate in the main manuscript to ensure the link can be made where it is required. Perhaps that would also have helped this reviewer find the relevant detail.

[5] In response to the reviewer comments we moved a description of the forcing and downscaling to the main text. With that gone, it appeared that the information that was previously in the Appendix could instead be provided in the main text without too much additional text, so we did that. We provided some more textual detail on the process descriptions. Including all energy balance equations would have to be added as a supplement, which did not seem to make much sense as the model equations are already documented online, as the editor points out.

It may be worth considering including the figure providing comparison against FLUXNET values in the supplementary material, in support of the comments in the main text. However, I would include in the main text a comment that you do not consider these as true validation I presume due to the issue of representation. This is commented on but maybe useful to add that additional sentence to clarify. Also, what is mean by N=16-168. Is this a typo?

[6] Agreed, we have taken the editor's advice and included more detail in the supplement, and also added an explicit statement along the lines suggested. We agreed the N=16-168 was confusing and changed it. It was not a typo as such, N=16 was for the study of Yebra, and N=168 for our unpublished analysis, which is now in the supplement (in revising this number increased to N=169 due to an additional site)

On the discussion on "before reaching ocean", and the need for better maps of closed basins. I would add to this that the reaching of the ocean is often also influence by evaporation (and the increase of evaporation due to irrigation). So while the DEM may indicate the river reaches the ocean, water in the river in fact does not. You could consider that not reaching the ocean is due to either being a closed basin or due to anthropogenic influences.

**[7] Agreed, see responses [51]-[53]**

I would like to suggest the authors update the manuscript based on the comments and the suggestions they have made in response. In that update please clearly outline the changes.

[8] Thank you. Below we outline the changes made. We also provide a copy of the m/s with the changes marked for the convenience of editor and reviewers.

**Response to Reviewer #1**

[9] We thank the reviewer for their comments and are glad that they enjoyed the m/s. We are also grateful for the editorial corrections and suggestions. Below we address the issues raised.

"The authors mention how all kind of modelling efforts will not produce independent and accurate estimates of irrigation water demand, but after the reading the objective I can only conclude that they will themselves do modelling as well. I think it would be good to refocus the introduction and state that models can have a valuable contribution but have their limitations and highlight how the authors would like to resolve these limitations."

[10] It is certainly true that our approach requires a model to assimilate the satellite observations into, and it is also true that additional assumptions are needed to translate water use estimates to irrigation water demand. We meant to emphasise that our method differs from existing methods in that it does not require mapping of the area irrigated or the extent of wetlands to estimate secondary evaporation. In revising, we added:

"Such an approach still involves modelling and the assumptions inherent to it, but the greater use of observations should mitigate against errors arising from the modelling." (I. 94-95)

**"I do not fully understand why this model is different from the other global water balance models out there. They authors should do a better job to highlight this, to emphasize why this model is better suited for this excursive than others"**

[11] The overall approach is different from existing models in that secondary evaporation is constrained by the satellite observations, rather than the result of simulation. If the reviewer refers to the W3RA model used in assimilation, then we believe a similar approach could be applied to other models, provided they have a coupled water and energy balance model and provided they are extended for data assimilation in the way described. We added:

"The W3RA model used here it not the only suitable modelling framework for the approach described. A similar method could be applied with other local or global models. The main requirements are that the model has a coupled water and energy balance model that simulates LST, and that it is amenable to data assimilation." (I. 168-171)

**"The quality of the forcing data is really low, how do the authors think this will impact the simulations and consequently the evaporation estimates?"**

[12] This question does not have a simple answer. The model takes several forcing data as input, the evaporation estimates are not equally sensitive to all of them, and the quality of forcing data also varies spatially and temporally

(seasonally as well as at longer time scales due to advances in satellite sensors). Hence it is impossible to give a quantitative answer to the question, but with regards to evaporation (only), one observed issue is that of heterogeneous biases in air temperature in regions with strong relief. Fortunately, secondary losses occur mostly in areas with low relief (see original text). To emphasise this more, we have added:

"A systematic bias in the global estimates of governing variables (radiation, air temperature and humidity, wind speed) are likely to be less problematic than spatially variable bias in these low-relief areas." (l. 596-598)

"Line 184-185 and Figure 1, I have the strong feeling that the model is biased in its estimates of E. Therefore, this would violate the basic assumption of a normal distribution with a mean of 0 around the observations. In addition, the authors cut-off the E' updates, which is in my opinion another violation of the EnKF. I feel the others should make sure that the model is bias free before implementing a DA technique like the EnKF. Otherwise, they can show the global biases to convince the reader that this is only the case for Figure 1, but I have to strong suspicion that it is also a problem for other regions (as for most models). I think the authors should address this large limitation in their discussion or somewhere else in the manuscript."

[13] Does the reviewer mean the model is biased in the absence of secondary evaporation (i.e. in drylands)? We do not have evidence for that. We have evaluated the ("offline" or background) model against evaporation rates reported by the global Fluxnet network for non-irrigated environments and did not find any bias, which is not surprising given the model was partly trained on those same data. We added the details of the evaluation, previously described as "unpublished", in a new supplement.

[14] The reviewer suggests showing global biases, but there are no global ET observations (other than Fluxnet) to calculate those from. However, we do compare with estimates from other models and find that ours are well within that range (see discussion in original text).

[15] We did not use EnKF but nudging based on energy balance model inversion.

[16] The reviewer is correct that we did cut off the E' updates. This was necessary to maintain internal physical consistency, but it is true that it may have introduced bias, particularly if the real E was consistently higher than the available energy, for example due to biases in meteorological forcing data. In revising the m/s, we have added:

"Values of the updated  $\lambda E'$  were constrained to positive values below or equal to potential evaporation EO, and therefore any gross underestimation of  $E_0$  by the model due to errors in meteorological forcing data would have resulted in an underestimation of the true evaporation rate." (I. 588-591)

"The manuscript could significantly benefit from a flowchart describing the full updating, calibration, nudging and assimilation procedure. Which variable are subject to what and where and how? The manuscript is difficult to follow without."

[17] Thank you for the suggestion. Data assimilation procedures are often difficult and tedious to explain but we have added a flow diagram to attempt to illustrate it better, and added the following explanation:

The methodology of our experiment includes two mostly separate components (Figure 1). The assimilation component integrates various MODIS products into the global hydrological model to estimate the dryland water balance and secondary evaporation. Subsequently, in an offline analysis the estimates of secondary evaporation were combined with mapping of irrigated crops to estimate a minimum irrigation requirement. Below follow details on the model, the data assimilation procedure, estimation of irrigation water use, and the different ways in which the results were evaluated. Details on the data used in the analysis can be found in the supplement to this article.

*Figure 1. Illustration showing the processing steps and data used in each step. Acronyms relate to input data that are described in the text.*

(l. 112-121)

"Line 253-254 Why is the increase in the estimation evaporation not from missing model processes? Incorrect vegetation parameterization or something else. This assumption is vital for the manuscript and is not really supported by argument on the model's quality to estimate evaporation in general. Has the model been validated against independent evaporation estimates?"

[18] This was discussed in the original m/s (l. 558-567 in the annotated revised m/s): the assimilation of satellite vegetation observations goes some way to address errors in vegetation parameterization. However, the (necessary) assumption that the assimilation increment is due to irrigation has uncertainty associated with if (but only if) most of a grid cell is occupied by non-irrigated land. Hence also the recommendation that our approach should work better at higher resolution, which we hope to pursue.

[19] Regards validation: see response [13].

"In addition, to the previous comment, the authors have not mention other forms of water use. I see no inclusion of domestic or industrial water use in the model nor in the estimates? Maybe these abstractions cause the errors in water basin closures."

[20] Domestic and industrial water use are not considered because these are typically non-consumptive uses (i.e., the water is returned to the environment after use). Possibly the main exception to this would be irrigation in urban landscapes, which the irrigation mapping does not capture well or at all. If those uses lead to surface cooling then the LST data assimilation will still have increased E estimates and so they are implicitly accounted for. In practice,

consumptive urban or industrial water uses are unlikely to have a meaningful impact on the water balance of large basins. We added:

"Domestic and industrial withdrawals are not considered here as a large fraction of these withdrawals is not evaporated but returned to the environment." (I. 391-393)

**"Line 129-134 are the calibrated parameter spatially consistent or are they really tuned to the individual basins?"**

[21] Neither, they vary spatially as a function of climate aridity and land cover using predictive relationships derived from model calibration to evaporation, soil moisture and streamflow from a very large number of sites and small and unregulated catchments, respectively. This was described in the original m/s, but we added a bit more detail in l. 155-162.

**"Line 134-135 Does the model have any lateral flow simulations of groundwater or surface water?"**

[22] No, only grid-based routing.

**"Line 150 a nudging factor of 0.99 is rather high, does this mean that the model is almost always wrong?"**

[23] Poor at predicting highly dynamic surface water extent, one could say, yes. (Like all global models, to the best of our knowledge.) We added:

"(reflecting the low skill in the model to accurate predict surface water extent at 0.05° resolution)" (l. 184-185)

**"Line 156-159 what is the spatial resolution of the Tair forcing, since it is very important for the LST simulations"**

[24] We agree that correct Tair is important, although the median bias correction step reduces most of the systematic difference, which we would argue is one of the novel aspects of our approach and one reason for its apparent success. These details were in the appendix, but we agree that they are probably important enough to be explained in the main text. Therefore we added:

"Monthly precipitation and air temperature climatology data at 30" from the WorldClim dataset (Hijmans et al., 2005) were resampled to 0.05° and 0.25°; subsequently, the ratio and difference, respectively, between the data at the finer and coarser resolution were applied to the forcing data." (I. 145-150)

**"Line 177 15degree, does this mean that the LST is spatially average over a 1500 by 1500km area???"**

[25] A 15x15° region is indeed about that size at lower latitudes. Note that this does not imply that LST is assumed homogenous across the area. This calculation is to remove the mean bias between daytime LST and time-of-overpass LST. We added *"to remove systematic bias"* (I. 213)

**"Line 508-510 the true error can also be larger. . . It is not said that it will be smaller due to the representativeness error."**

[26] In theory, yes, although given the rather large sample such a statistical accident would be unlikely. Nonetheless we removed this statement, for the more important reason that the relatively large uncertainty in Fluxnet energy balance terms means that they do not provide a very reliable assessment of possible bias in our model estimates (see new supplement).

"Line 581-583 As far as I understand most other models use sub-grid parameterization, which would allow for a partial coverage of the grid cell by irrigation areas. This statement is therefore potentially incorrect and should be removed to avoid misinforming the reader"

[27] We respectfully disagree. The MIRCA2000 mapping suggests the grid cell is 100% equipped for irrigation. To our knowledge the published models assume that the entire equipped area is irrigated so the statement holds. Of course that assumption could be changed for another.

"Line 619-623 I feel the units are incorrect, I guess the first estimates should be 75.5\*1012 Km3 y-1 (as well as for the other estimates from this study, which are now 1000 times lower than other studies)

[28] The units are correct. We could have written 75,500 km3 y-1 but felt using base units (m) was more appropriate, as neither unit is easily imagined.

**Response to Reviewer 2**

[29] We thank the reviewer for their positive and constructive comments. Below we respond to the issues raised.

**"First of all, I find the manuscript a bit unbalanced in terms of contents. There is a lot of focus on methods and equations (esp. for irrigation), but relatively a few figures for results. This makes the manuscript very tedious to read with a lot of text and information. At the same time, some information that are critical to assess the results are either missing or in the appendix. For example, forcings and their spatial disaggregation, model formulations of LE and H, etc."**

[30] We are sorry the m/s was tedious to read. We appreciate that the technical detail of the modelling and data assimilation can be a bit tedious, which is why we tried to minimise that aspect in the body text by transferring some of the material to the appendix and referring to existing studies where possible. We have added 2 figures: one illustrating the workflow, and one with some new analysis suggested by the reviewer. We hope this has made the m/s less tedious.

[31] The referee also asks for additional material to be included and we therefore made some additions. We have added further details on "forcings and their spatial disaggregation" in the main text (see response [24]). The "model formulations of LE and H" were described in the methods section. The energy balance equation is the main model component of relevance here, but it is in essence a conventional implementation of the Penman-Monteith equation, which is well-known and the detail of the implementation is readily available online already. We did some additional text to explaining the approach however:

"The surface energy and water balance is simulated using the Penman-Monteith model. The evaporative fluxes from transpiration, unsaturated soil, saturated soil and surface water are simulated subject to the overall constraint of potential evaporation EO within the same Penman-Monteith framework. Wet canopy evaporation is simulated outside this constraint, for reasons described in Van Dijk et al. (2015), using a dynamic-canopy version of the event-based Gash model (Van Dijk and Bruijnzeel, 2001; Wallace et al., 2013)." (I. 133-138)

**"Definition of the secondary evaporation: There is no description on how groundwater's contribution to LE/ET is a secondary source. In an idealistic theoretical situation, the capillary flux from groundwater will replenish soil moisture (at some point when the soil moisture is drying up), which would eventually increase LE. It is not clear if the model considers such capillary flux processes explicitly. I am curious about what fraction of 'other' sources is actually coming from groundwater-soil-LE pathway, and not groundwater-baseflow-surface water-LE pathway. The first one may have a critical influence on vegetation and carbon cycle processes."**

[32] The model does consider capillary fluxes, but in the offline model those are ultimately constrained by longerterm local rainfall, and therefore do not constitute secondary evaporation (i.e., it is derived from locally recharged, unconfined groundwater rather than lateral groundwater inflows). As our study demonstrates, data assimilation helps to estimate secondary evaporation from non-local water sources, but does not directly attribute it to a water source – that requires ancillary data. In some cases, the secondary evaporation may be from irrigation with water pumped from confined aquifers (which bypasses the capillary rise pathway). In other cases, it is possible that secondary evaporation is inferred, e.g. because rainfall is underestimated, capillary rise or deep root water uptake is more important than predicted by the background model (e.g., because the vegetation is more deeply rooted or groundwater is closer to the surface than assumed). There is obviously much more to be done to understand the global water balance in full detail. Our data provide a means of prioritising regions where there appears to be hydrological behaviour that is not easily explained by the background model, and therefore is worthy of further investigation. To make clear that capillary rise is possible within the model, we added the following words:

"[The soil column is conceptualised as a three-layer unsaturated zone overlaying an unconfined groundwater store], *from which capillary rise can occur.*" (I. 130-131)

**"Assimilation of LST into model: In the assimilation of LST into model, the basic assumption is that the modelsimulated partitioning of the energy fluxes (H and E) are correct. The corrections or 'nudges' for LST are backcalculated from the modelled H, and these are propagated through spatial patterns of observed LST. But, there is no explanation of how 'background' H and LE are calculated in the model. Perhaps, these may be inferred from previous papers/reports on the model (?), but they are so critical for this study and results presented herein, they deserve to be in this manuscript."**

[33] The basic assumption is actually not that the partitioning of H and LE is correct, but rather, that the estimated total available energy (A=H+LE) is correct. Data assimilation may change the estimate of H, and through that LE=A-H. To make this clear we added:

"A fundamental assumption in this approach is that the partitioning between  $\lambda E$  and H can be improved with information on LST, but that the estimate of available energy A is correct." (I. 224-226)

[34] The background H and LE are estimated using the conventional Penman-Monteith approach. We have added new details on that in the model description section (see [31]).

*# "One information that is imperative is whether the parameters of the modelled LE and H were optimized or not. If not, are the used parameter values are reasonable for a global-scale application?"*

[35] The most important parameter overall, surface conductance, was predicted from satellite-observed surface reflectances following Yebra et al. (2013) and tuned using a large data base of evaporation measurements (FLUXNET). Another important parameter, vegetation height (affecting aerodynamic conductance) was derived from remote sensing by Simard et al. (2011). We have added a few additional words to hopefully make the approach clearer:

"Global datasets were also used to parameterise the distribution of different land surface types (Bicheron et al., 2008) and the properties of vegetation (Simard et al., 2011), soil (Shangguan et al., 2014), and aquifers (Gleeson et al., 2014; Beck et al., 2015)." (I. 150-152)

and

"[Five model parameters that were both relatively uncertain and influential were calibrated and regionalised] *by climate and land cover type class,* [using large global data sets of site measurements evaporation and near-surface soil moisture, and a global dataset of catchment streamflow records (the parameters represent proportional adjustments to initial estimates of, respectively, maximum canopy conductance, relative canopy rainfall evaporation rate, soil evaporation, saturated soil conductivity, and soil conductivity decay with depth).]" (I. 1487-153)

**"Related to the above point, validation for model simulated LE and H is not shown or discussed. There are references to a previous study or an unpublished work but the findings of this study also warrant a section on evaluations at the global scale. I am aware that observed global ET and H data are not available, but a comparison with either FLUXNET observations (for sites) or other satellite-based ET products can provide a valuable benchmark."**

[36] In response, we have summarised the result of the unpublished evaluation and included it as a new supplement. We believe putting it in the main text would be misleading readers into thinking it constitutes an assessment of the performance in estimating secondary evaporation, which it does not: the vast majority of flux towers are in environments without secondary evaporation. This was also the reason we initially did not think it a good idea to include it, but we can see that a reader might want to see anything that is referred to and that "unpublished'

therefore might not cut it. As the supplement makes clear, the flux tower observations also suffer from the energy balance closure problem which makes evaluation more ambiguous.

[37] A comparison with other global ET products was discussed in the original m/s.

**"Estimation of irrigation water use: Assumption of rooting depth: The parameter smax is dependent on the assumed rooting depth. The manuscript would benefit from a discussion on how these parameters vary globally, and to what extent do this variation affects the estimation of secondary evaporation from irrigated area."**

[38] We follow the published methodology of Siebert and Döll (2010). The assumptions made here do not affect the estimation of secondary evaporation. They do affect is the calculated irrigation efficiency and therefore the estimate of irrigation water use. This is a perhaps subtle, but important distinction. We added additional text in to places to make sure this is clear:

"The assimilation component integrates various MODIS products into the global hydrological model to estimate the dryland water balance and secondary evaporation. Subsequently, in an offline analysis the estimates of secondary evaporation were combined with mapping of irrigated crops to estimate a minimum irrigation requirement." (I. 112-116)

and

"The estimation of  $I_0$  was done after, and entirely separate from, the data assimilation process, and therefore what follows had no bearing on the estimation of secondary evaporation." (I. 267-269)

*# "Evaluation against discharge observations: In my subjective judgment, the improvement in the basins with discharge